# IFITM proteins promote SARS-CoV-2 infection and are targets for virus inhibition in vitro

Caterina Prelli Bozzo[1,13], Rayhane Nchioua[1,13], Meta Volcic [1], Lennart Koepke [1], Jana Krüger [2], Desiree Schütz[1], Sandra Heller [2], Christina M. Stürzel[1], Dorota Kmiec [1,3], Carina Conzelmann[1], Janis Müller [1], Fabian Zech[1], Elisabeth Braun[1], Rüdiger Groß [1], Lukas Wettstein [1], Tatjana Weil [1], Johanna Weiß[1], Federica Diofano[4], Armando A. Rodríguez Alfonso [5,6], Sebastian Wiese [6], Daniel Sauter [1,7], Jan Münch [1], Christine Goffinet [8], Alberto Catanese [9], Michael Schön[9], Tobias M. Boeckers [9,10], Steffen Stenger[11], Kei Sato [12], Steffen Just [4], Alexander Kleger [2], Konstantin M. J. Sparrer [1✉] & Frank Kirchhoff [1✉]

Interferon-induced transmembrane proteins (IFITMs 1, 2 and 3) can restrict viral pathogens, but pro- and anti-viral activities have been reported for coronaviruses. Here, we show that artificial overexpression of IFITMs blocks SARS-CoV-2 infection. However, endogenous IFITM expression supports efficient infection of SARS-CoV-2 in human lung cells. Our results indicate that the SARS-CoV-2 Spike protein interacts with IFITMs and hijacks them for efficient viral infection. IFITM proteins were expressed and further induced by interferons in human lung, gut, heart and brain cells. IFITM-derived peptides and targeting antibodies inhibit SARS-CoV-2 entry and replication in human lung cells, cardiomyocytes and gut organoids. Our results show that IFITM proteins are cofactors for efficient SARS-CoV-2 infection of human cell types representing in vivo targets for viral transmission, dissemination and pathogenesis and are potential targets for therapeutic approaches.

[1] Institute of Molecular Virology, Ulm University Medical Center, Ulm, Germany. [2] Department of Internal Medicine I, Ulm University Medical Center, Ulm, Germany. [3] Department of Infectious Diseases, King's College London, London, UK. [4] Department of Internal Medicine II (Cardiology), Ulm University, Ulm, Germany. [5] Core Facility of Functional Peptidomics, Ulm University Medical Center, Ulm, Germany. [6] Core Unit of Mass Spectrometry and Proteomics, Ulm University Medical Center, Ulm, Germany. [7] Institute of Medical Virology and Epidemiology of Viral Diseases, University Hospital Tübingen, Tübingen, Germany. [8] Institute of Virology, Charité—Universitätsmedizin Berlin, Berlin, Germany. [9] Institute for Anatomy and Cell Biology, Ulm University, Ulm, Germany. [10] Deutsches Zentrum für Neurodegenerative Erkrankungen (DZNE), Ulm University, Ulm, Germany. [11] Institute of Medical Microbiology and Hygiene, Ulm University Medical Center, Ulm, Germany. [12] Institute of Medical Science, The University of Tokyo, Tokyo, Japan. [13] These authors contributed equally: Caterina Prelli Bozzo, Rayhane Nchioua. ✉email: Konstantin.Sparrer@uni-ulm.de; Frank.Kirchhoff@uni-ulm.de

SARS-CoV-2 is the cause of the Coronavirus disease 2019 (COVID-19) pandemic. The first cases were described in China in late 2019. Since then, the virus has infected more than 170 million people around the globe (https://coronavirus.jhu.edu/map.html, June 2021). While SARS-CoV-2 spreads more efficiently than severe acute respiratory syndrome coronavirus (SARS-CoV) and the Middle East Respiratory Syndrome coronavirus (MERS-CoV), which previously emerged, it has a lower case-fatality rate (~2–5%), compared to ~10% and almost 40%, respectively[1–3]. The reasons for this efficient spread and the mechanisms underlying the development of severe COVID-19 are incompletely understood but the ability of SARS-CoV-2 to evade or counteract innate immune mechanisms may play a key role[4,5].

Here, we focused on innate immune effectors that are thought to target the first essential step of SARS-CoV-2 replication: entry into its target cells. A prominent family of interferon (IFN) stimulated genes (ISGs) known to inhibit fusion between the viral and cellular membranes are interferon-inducible transmembrane (IFITM) proteins[6,7]. The three best-characterized members of the IFITM family are IFITM1, IFITM2, and IFITM3[8–11]. They contain different sorting motifs and IFITM1 is mainly localized at the plasma membrane, while IFITM2 and 3 are found inside the cell on endo-lysosomal membranes[8]. Thus, IFITM proteins may act at different sites of viral entry and it has been reported that they restrict multiple classes of enveloped viral pathogens including Influenza A viruses, Flaviviruses, Rhabdoviruses, Bunyaviruses, and human immunodeficiency viruses[7,12]. The molecular mechanism(s) underlying the antiviral activity of IFITMs are not fully understood. However, recent reports suggest that they modulate membrane rigidity and curvature to prevent fusion of the viral and cellular membranes[13–17].

It has also been reported that IFITM proteins inhibit human coronaviruses including SARS-CoV-1 and SARS-CoV-2 as well as MERS-CoV[10,12,18–21]. However, most results were obtained using Spike containing viral pseudo particles and cell lines overexpressing the IFITM proteins and frequently also the viral ACE2 receptor. Here, we confirmed and expanded previous results showing that IFITM proteins block SARS-CoV-2 entry under such artificial experimental conditions. In contrast, however, endogenous expression of IFTIM proteins was essential for efficient infection and replication of genuine SARS-CoV-2 in human cell types involved in virus transmission, dissemination to various organs, and development of severe COVID-19. In further support of an important role of IFITM proteins as entry cofactors of SARS-CoV-2, IFITM-derived peptides and targeting antibodies efficiently inhibited SARS-CoV-2 infection of human lung, heart, and gut cells. Our unexpected finding that SARS-CoV-2 hijacks human IFITM proteins for efficient infection helps to explain the rapid spread of this pandemic viral pathogen.

## Results

### Overexpressed IFITMs block but endogenous IFITMs boost SARS-CoV-2 infection

It has been reported that overexpression of IFITM proteins prevents entry of viral particles pseudotyped with the Spike (S) proteins of SARS- and MERS-CoVs[10,12,18,19]. In agreement with these previous results, we found that IFITM1, IFITM2, and (less efficiently) IFITM3 dose-dependently inhibit SARS-CoV-2 S-mediated entry of Vesicular-Stomatitis-Virus pseudo particles (VSVpp) into transfected HEK293T cells (Fig. 1a, Supplementary Fig. 1a, b). Control experiments showed that the IFITM1 and IFITM3 antibodies used were specific, while the IFITM2 antibody also recognized IFITM3, albeit less efficiently (Supplementary Fig. 1c). Inhibition of SARS-CoV-2 S-mediated infection by IFITM proteins was confirmed using

lentiviral pseudo particles (LVpp, Supplementary Fig. 1d). In contrast, IFITMs did not significantly affect VSV-G-dependent entry (Supplementary Fig. 1e, Supplementary Data 1). It has been reported, that IFITMs can be incorporated into budding HIV-1 virions and reduce their infectivity[22,23]. Thus, we also examined whether IFITMs expression in the virus producer cells affects SARS-CoV-2 S VSVpp production (Supplementary Fig. 2a). However, overexpression of IFITMs in the producer cells had little if any restrictive effects (Supplementary Fig. 2b). IFITM1 even slightly increased VSVpp infectivity at low expression levels, possibly because it moderately enhances incorporation of SARS-CoV-2 S into VSV particles (Supplementary Fig. 2c).

To examine the impact of endogenous IFITM expression on S-mediated VSVpp infection, we performed siRNA knockdown (KD) studies in the human epithelial lung cancer cell line Calu-3, which expresses ACE2[24] and increasing levels of IFITM proteins upon IFN treatment (Supplementary Fig. 3a). Quantitative polymerase chain reaction (qPCR) analysis using primers that are specific for the three IFITM genes (Supplementary Fig. 3b, Supplementary Table 1), showed that the siRNAs mainly silenced their respective IFITM mRNA targets (Supplementary Fig. 3c), although cross-silencing by IFITM2 and IFITM3 siRNAs was observed in some experiments. This did not come as surprise since IFITM2 and IFITM3 are highly homologous. On average, silencing of IFITM expression enhanced VSVpp infection mediated by SARS-CoV S proteins about three- to seven-fold (Fig. 1b). To determine whether overexpression of IFITMs also affects genuine SARS-CoV-2 replication, we infected HEK293T cells overexpressing ACE2 alone or together with individual IFITM proteins. In agreement with the inhibitory effects on S containing VSVpp and LVpp, IFITM1 and IFITM2 prevented viral RNA production almost entirely, while IFITM3 achieved approximately five-fold inhibition (Fig. 1c).

To further approximate the in vivo situation, we examined the role of endogenous IFITM expression on genuine SARS-CoV-2 infection of human lung cells. In contrast to the results obtained with pseudovirions and/or IFITM overexpression, silencing of endogenous IFITM expression in Calu-3 cells strongly impaired viral RNA production (Fig. 1d, Supplementary Fig. 4a–c, Supplementary Table 1). On average, IFITM2 knockdown reduced viral RNA yields by ~20-fold in the absence and by~68-fold in the presence of IFN-β. Consequently, the amount of infectious SARS-CoV-2 particles in the cell culture supernatant was reduced by several orders of magnitude upon silencing of IFITM2 and to a lesser extent also by depletion of IFITM1 and IFITM3 as assessed by both TCID50 (50% Tissue Culture Infectious Dose) and plaque assays (Fig. 1e–g, Supplementary Fig. S4d, e). There was a highly significant correlation ($r = 0.82$ for TCID50 and $r = 0.83$ for plaque assays) between the infectious viral particles and viral RNA levels in the supernatants (Fig. 1h, i), indicating that the qPCR accurately represents infectious virus particle yield but may even underestimate the effects[5,24]. Titration analyses showed that IFITMs do not promote SARS-CoV-2 infection in transfected HEK239T cells over a broad range of expression levels (Supplementary Fig. 5). Thus, the opposing effects of transient and endogenous IFITM expression were not just due to different expression levels. Altogether, these results showed that endogenous IFITMs promote rather than prevent SARS-CoV-2 infection.

### IFITMs enhance SARS-CoV-2 infection in primary human lung cells

To confirm that the requirement of endogenous IFITM expression for efficient SARS-CoV-2 replication is not limited to Calu-3 cells, we silenced IFITM proteins in primary small airway epithelial cells (SAEC) isolated from normal human lung tissues. Western blot analyses showed that SAEC cells express all three

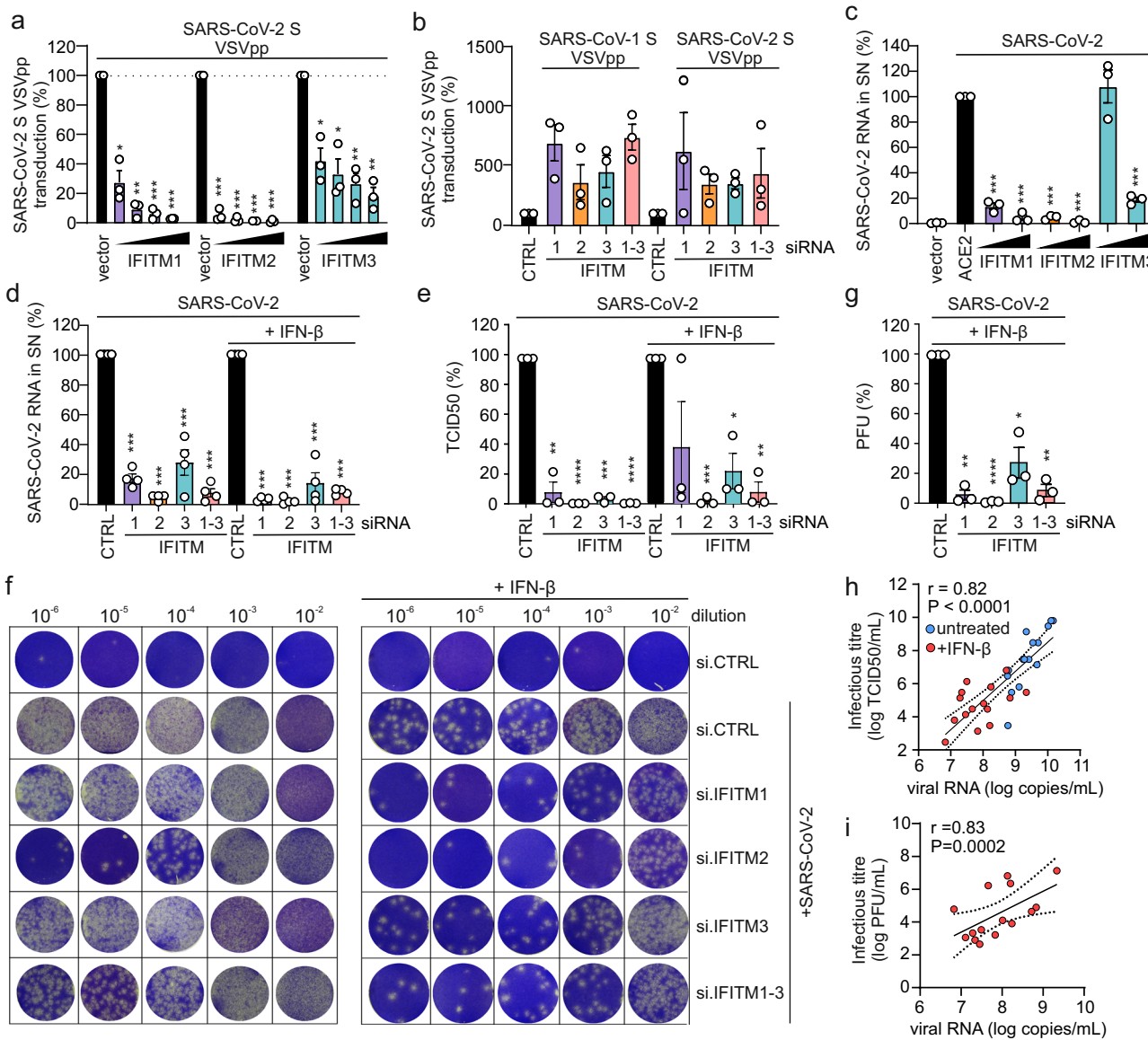

**Fig. 1 Opposing effects of IFITM proteins on SARS-CoV-2 infection. a** Quantification of VSV(luc)ΔG*SARS-CoV-2-S entry by measuring luciferase activity in HEK293T cells transiently expressing the indicated IFITM proteins. Bars represent the mean of three independent experiments (±SEM), exact p values are provided in supplementary data 1. **b** Calu-3 cells treated with non-targeting (CTRL) or IFITM1, 2, or 3 siRNAs or a combination of the three and infected with VSV(luc)ΔG*SARS-CoV-2-S particles. Bars represent the mean of three independent experiments (±SEM). **c** Quantification of viral N RNA levels by qRT-PCR 48 h post-infection with SARS-CoV-2 (MOI 0.05) in the supernatant of HEK293T cells transiently expressing ACE2 alone or together with the indicated IFITM proteins. Bars represent the mean of three independent experiments (±SEM), exact p values are provided in the supplementary data 1. **d** Quantification of viral N RNA levels by qRT-PCR in the supernatant of Calu-3 cells, collected 48 h post-infection with SARS-CoV-2 (MOI 0.05). Cells were transfected with control (CTRL) or IFITM1, 2, and/or 3 targeting siRNA or a combination of the three and either treated with IFN-β or left untreated as indicated. Bars represent the mean of four independent experiments (±SEM). ***p < 0.0001. **e** Infectious SARS-CoV-2 particles in the supernatant of (**d**) as assessed by TCID50 assay. Bars represent the mean of three independent experiments (±SEM), exact p values are provided in Supplementary Data 1. **f** Infectious SARS-CoV-2 particles in the supernatant of (**d**) as assessed by plaque-forming unit assay. **g** Quantification of (**f**). Bars represent the mean of three independent experiments (±SEM). siRNA CTRL vs. siRNA IFITM1 p = **0.0012, siRNA CTRL vs. siRNA IFITM2 p = ****<0.0001, siRNA CTRL vs. siRNA IFITM3 p = *0.0197, siRNA CTRL vs. siRNA IFITM1–3 p = **0.0020). **h, i** Correlation of infectious viral particle analysis (**h**, TCID50; **i**, plaques) of (**e**, **g**) with qPCR data from (**d**) values showed calculated as a Pearson correlation (**h**: r = 0.82, p < 0.0001, **i**: r = 0.83, p = 0.0002). **a**, **c**, **d**, **g** Unpaired t test with Welch's correction. **b**, **e** Unpaired t tests. p Values are indicated as *p < 0.05; **p < 0.01; ***p < 0.001.

IFITM proteins and type I IFN treatment efficiently enhanced the expression levels while IFN-γ had only modest effects (Fig. 2a). siRNA-mediated silencing was associated with strongly reduced levels of SARS-CoV-2 RNA production in the presence of IFN-β (Fig. 2b, Supplementary Fig. 6a). In agreement with inefficient induction by type II IFN (Fig. 2a), kd of IFITMs did not affect viral RNA yields upon treatment with IFN-γ (Fig. 2c, Supplementary

Fig. 6b). Notably, silencing of IFITM1 also clearly reduced viral RNA yields in the absence of IFN treatment (Fig. 2b, c). Altogether, IFITM1 was more critical for efficient SARS-CoV-2 replication in SAEC cells than in Calu-3 cells. It is thought that IFITM1 is mainly found at the cell surface, while IFITM2 is preferentially localized in early endosomes[7,8]. SARS-CoV-2 may enter cells at their surface as well as in endosomes[25]. Thus, different expression levels of IFITM

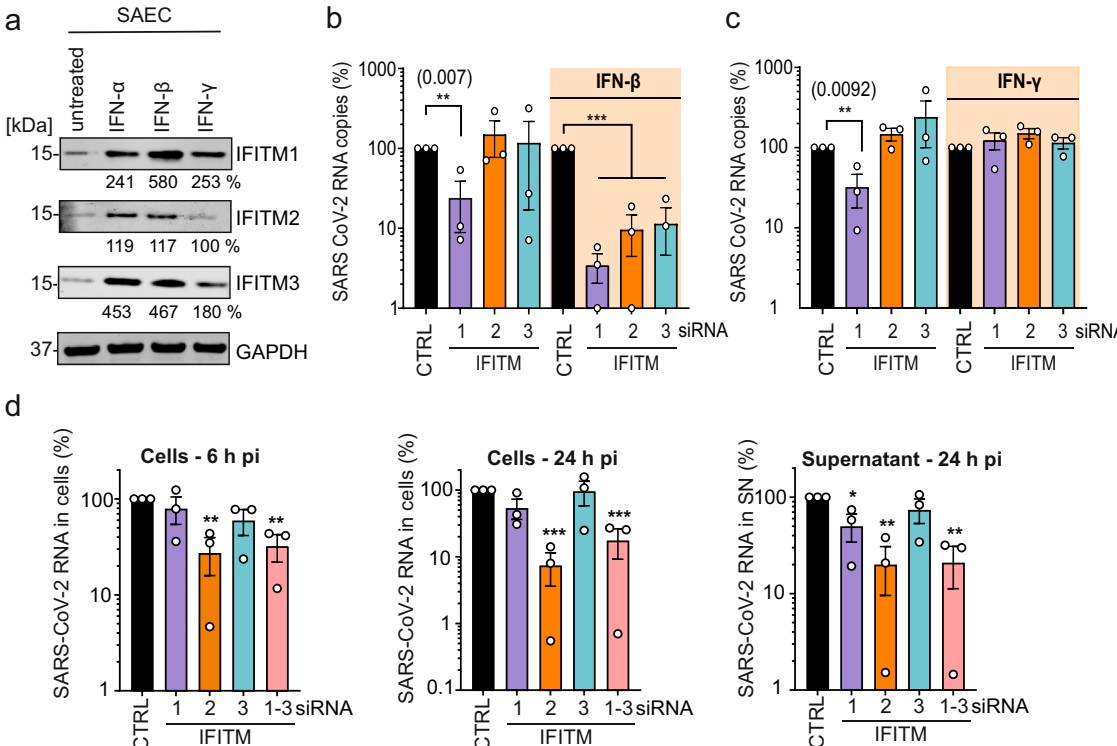

**Fig. 2 Role of IFITMs in SARS-CoV-2 replication in SAEC. a** Expression of IFITM1, IFITM2, and IFITM3 in SAEC after stimulation with IFN-α2 (500 U/ml, 72 h), IFN-β (500 U/ml, 72 h), or IFN-γ (200 U/ml, 72 h). Immunoblots of whole-cell lysates were stained with α-IFITM1, α-IFITM2, α-IFITM3, and α-GAPDH. The experiment was performed twice to similar results. **b, c** Quantification of SARS-CoV-2N RNA in the supernatants of SAEC that were left untreated or treated with IFN-β (**b**) or IFN-γ (**c**) 48 h post-infection with SARS-CoV-2 (MOI 2.5). Shown are viral RNA levels relative to those obtained upon treatment with the control siRNA (100%). Bars represent the means of three independent experiments (±SEM). (**b**, untreated panel) siRNA CTRL vs. siRNA IFITM1 $p =$ **0.007; (**b**, IFN-β treated panel) siRNA CTRL vs. siRNA IFITM1/2/3 $p =$ ***0.0001; (**c**, untreated panel) siRNA CTRL vs. siRNA IFITM1 $p =$ ** 0.0092. **d** Quantification of intracellular viral N RNA levels in Calu-3 cells 6 h (left panel) and 24 h (middle panel) post-infection with SARS-CoV-2 (MOI 0.05). (right panel) Quantification of viral N RNA levels in the supernatant 24 h post-infection. Values were normalized to GAPDH and calculated relative to the control (set to 100%). Cells were transiently transfected with siRNA either control (CTRL) or targeting IFITM1, 2, 3, or a combination of the three as indicated. Bars represent the mean of three independent experiments, measured in duplicates (±SEM), exact $p$ values are provided in Supplementary Data 1. **d** Unpaired $t$ test with Welch's correction. **b, c** Unpaired $t$ tests. $p$ Values are indicated as *$p < 0.05$; **$p < 0.01$; ***$p < 0.001$ or were not significant ($p > 0.05$).

proteins, as well as cell-type-dependent differences in the major sites of viral fusion, may explain differences in the relative dependency of SARS-CoV-2 on endogenous IFITM1 or IFITM2/3 expression. In contrast to the results obtained in Calu-3 cells (Supplementary Fig. 4c), IFN-β enhanced rather than inhibited SARS-CoV-2 replication in SAEC cells (Supplementary Fig. 6a). While this finding came as surprise, it is reminiscent of previous data showing that IFN treatment promotes infection by human coronavirus HCoV-OC43. Notably, this CoV was proposed to hijack IFITM3 for efficient entry[26]. Taken together, our results show that endogenous expression of IFITM proteins promotes SARS-CoV-2 replication in primary human lung cells, especially in the presence of IFN-β.

**Endogenous IFITMs promote an early step of SARS-CoV-2 infection**. To address the mechanisms underlying these opposing effects of IFITMs, we examined the effect of IFITM proteins on SARS-CoV-2 S-mediated fusion under various conditions. To analyze the impact of IFITMs on S-mediated fusion between virions and target cells, we used HIV-1 particles containing β-lactamase-Vpr fusions[27], except that the virions contained the SARS-CoV-2 S instead of the HIV-1 Env protein. In agreement with the documented role of IFITMs as inhibitors of viral fusion[13,15], transient overexpression of all three IFITM proteins blocked fusion of SARS-CoV-2 S HIVpp[27] with ACE2 expressing

HEK293T cells (Supplementary Fig. 7a). Consistent with recent data[20] results from a split-GFP assay showed that artificial overexpression of IFITMs also inhibits HEK293T cell-to-cell fusion mediated by the SARS-CoV-2 S protein and the ACE2 receptor (Supplementary Fig. 7b). To analyze the impact of endogenous IFITM expression on genuine SARS-CoV-2 entry, we determined the levels of viral RNA in the cells at different time points after infection of Calu-3 cells. Already at 6 h post-infection, depletion of IFITM2 significantly reduced the intracellular levels of viral RNA (Fig. 2d). At 24 h post-infection, silencing of IFITM2 expression decreased intracellular SARS-CoV-2 RNA levels by ~15-fold and extracellular viral RNA yield by ~5-fold (Fig. 2d). These results support that endogenous IFITM2 expression promotes an early step of SARS-CoV-2 infection of human lung cells.

**Evidence for SARS-CoV-2 Spike interaction with IFITM proteins**. It is thought that the broad-spectrum antiviral activity of IFITM proteins does not involve specific interactions with viral proteins but affects the properties of cellular membranes[6,8,28]. To assess whether the ability of SARS-CoV-2 to utilize IFITMs for efficient infection of human lung cells may instead involve specific interactions between the viral S protein and IFITMs, we performed proximity ligation assays (PLA; Supplementary Fig. 8a)[29]. The results revealed a higher number of PLA foci for S and IFITM2 compared to

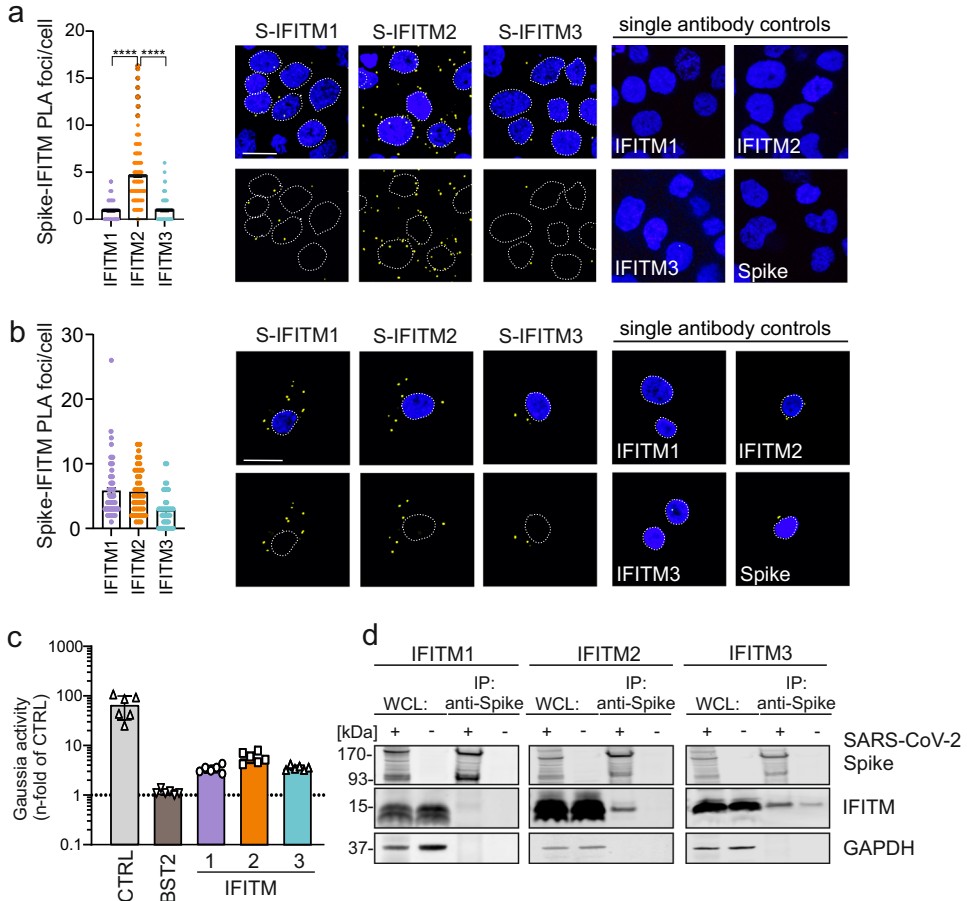

**Fig. 3 IFITM2 promotes SARS-CoV-2 entry and interacts with the Spike protein. a** Quantification and exemplary images of a proximity ligation assay (PLA) between the SARS-CoV-2 Spike and IFITM proteins in Calu-3 cells infected with SARS-CoV-2 for 2 h at 4 °C. DAPI (blue), nuclei. PLA signal (yellow), proximity between S/IFITMs. Bars represent the mean of three independent experiments, (300 cells, ±SEM), two-sided Wilcoxon matched-pairs test, ****$p < 0.0001$. Scale bar, 20 μm. **b** Quantification and exemplary images of a PLA in SAEC using a similar setup as (**a**). DAPI (blue), nuclei. PLA signal (yellow), proximity between S/IFITMs. Scale bar, 20 μm. Bars represent the mean of two independent experiments (80 cells, ±SEM). **c** Relative interaction between SARS-CoV-2 Spike and human IFITM proteins measured by MaMTH protein-protein interaction assay in cotransfected HEK293T B0166 *Gaussia* luciferase reporter cells. Bars represent the mean of triplicate transfections performed in two independent experiments. **d** Co-Immunoprecipitation (Co-IP) of IFITM proteins by the Spike protein in HEK293T cells overexpressing SARS-CoV-2 S and IFITM1, IFITM2, or IFITM3, as indicated. Twenty-four-hours post-transfection, cells were harvested. The Co-IP was repeated twice to similar results.

IFITM1 and 3 in SARS-CoV-2 infected Calu-3 lung cells (Fig. 3a), indicating close proximity of these two proteins. In accordance with the relevance of IFITM1 for SARS-CoV-2 replication in this cell type, high levels of PLA signals were detected for S and IFITM1 in infected primary human SAEC cells (Fig. 3b). Mutation of Y19D reported impairing endocytosis of IFITM2[30] significantly reduced the number of Spike-IFITM2 PLA signals in SARS-CoV-2 infected Hela-ACE2 cells (Supplementary Fig. 8b) suggesting that proper IFITM localization is required for interaction. Notably, the results underestimate the actual effect of the Y19D mutations on the interaction since Hela-ACE2 cells express IFITM2 endogenously. Examination of integral membrane protein-protein interactions using the mammalian-membrane two-hybrid (MaMTH) assay[31] provided further evidence that SARS-CoV-2 S interacts with IFITM proteins. In brief, the interaction of membrane proteins leads to the release of transcription factors that activates the expression of the Gaussia luciferase reporter gene (Supplementary Fig. 9). All IFITM proteins resulted in significantly higher levels of luciferase activity compared to the BST2 (tetherin) control construct with IFITM2 having the highest signal activity (Fig. 3c). Finally, the SARS-CoV-2 S-protein co-immunoprecipitated IFITM2 and, to a lesser extent, IFITM1 and IFITM3 (Fig. 3d). Altogether, several independent lines of evidence support

that the S protein of SARS-CoV-2 interacts with human IFITM proteins, especially with IFITM2.

**Effects of endogenous IFITM expression in human lung cells on Spike-ACE2 interaction.** Next, we examined whether IFITMs affect the interaction between the SARS-CoV-2 S protein and the ACE2 receptor. KD of IFITM2 and, to a lesser extent, IFITM3 enhanced the number of S/ACE2 PLA foci after infection of Calu-3 cells with genuine SARS-CoV-2 (Fig. 4a). The number of S/ACE2 foci rapidly declined (Fig. 4b) and S/RAB5A signals strongly increased after switching SARS-CoV-2 infected Calu-3 cell cultures from ice to 37 °C (Fig. 4c). Upon silencing of IFITM2 these effects were less pronounced (Fig. 4d) and the number of S molecules that are in close proximity to the ACE2 receptor decreased. Thus, IFITM2 may promote SARS-CoV-2 S interaction with ACE2 in early endosomes. In agreement with this, S/IFITM2 PLA foci rapidly increased after brief incubation of SARS-CoV-2 infected Calu-3 cell cultures at 37 °C and colocalized with the early endosome marker EEA1 but hardly with the late endosome marker RAB7a (Fig. 4e–g, Supplementary Movies 1 and 2). Altogether, these results indicate that IFITM2 may already interact

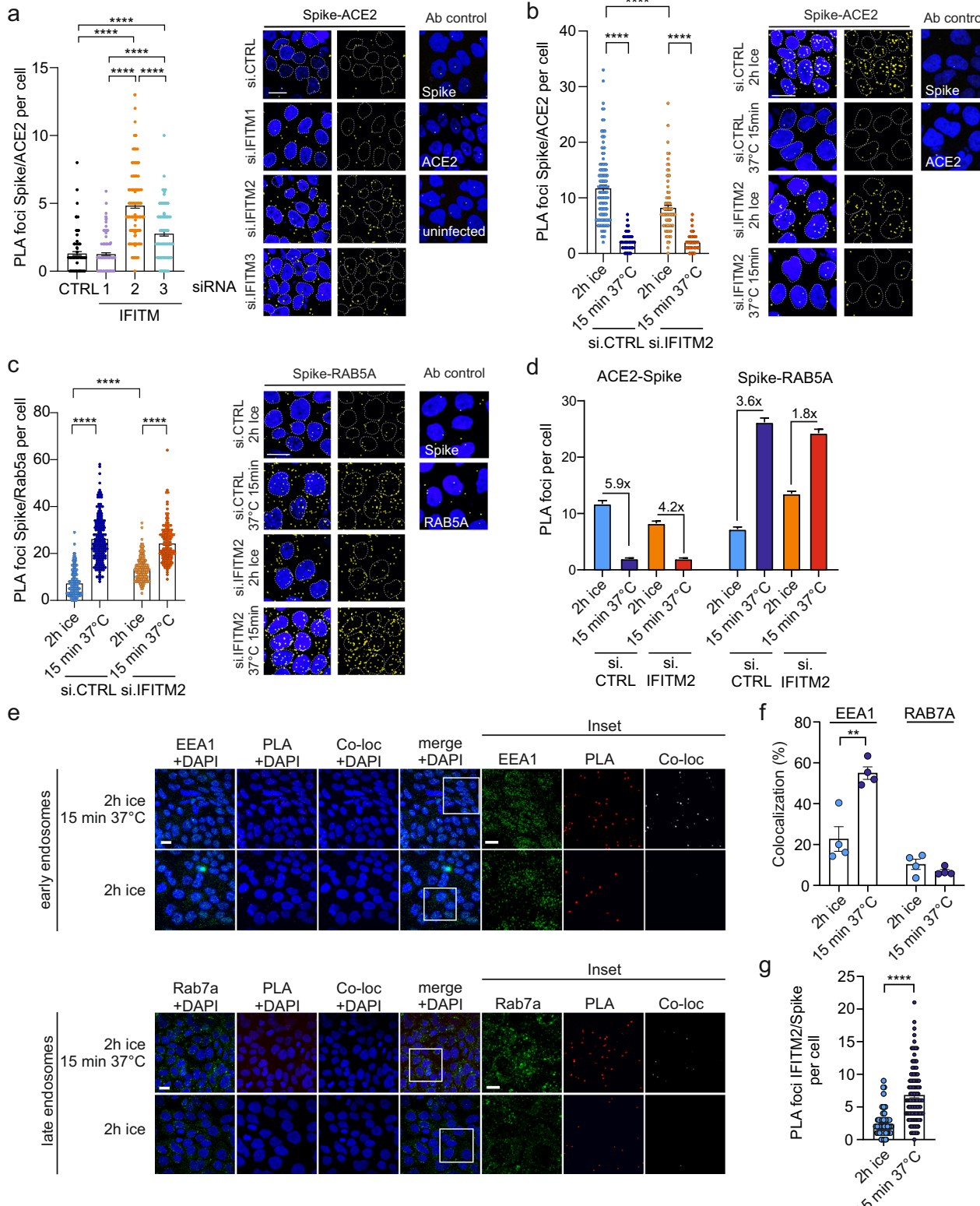

**IFITMs are targets for inhibition of SARS-CoV-2 replication.** Our discovery that IFITMs serve as cofactors for efficient SARS-CoV-2 infection suggested that they might represent targets for viral inhibition. To address this, we examined the effect of

with the SARS-CoV-2 S at the cell surface and promotes subsequent virus-cell fusion predominantly in early endosomes.

antibodies targeting the N-terminal region of the three IFITM proteins (Fig. 5a) on SARS-CoV-2 infection of Calu-3 cells. Indeed, antibodies against the N-terminal region of IFITM2 or recognizing all three IFITM proteins inhibited SARS-CoV-2 replication in Calu-3 cells up to 50-fold, while antibodies against IFITM1 or IFITM3 had negligible inhibitory effects (Fig. 5b). Infection experiments showed that peptides corresponding to the N-proximal region of IFITM2 that is recognized by the SARS-

**Fig. 4 Impact of IFITMs on the ACE2-SARS-CoV-2 S proximity. a** Quantification and exemplary images of a PLA between SARS-CoV-2 Spike and ACE2 in Calu-3 depleted of IFITM1, IFITM2, or IFITM3 and infected with genuine SARS-CoV-2. Bars represent the mean of four independent experiments (100 cells ±SEM), two-sided Wilcoxon matched-pairs test, ****$p < 0.0001$. **b** Quantification and exemplary images of a PLA between Spike and ACE2 in Calu-3 cells depleted of IFITM2 and infected with SARS-CoV-2 virus on ice for 2 h and then incubated for 15 min at 37 °C. Bars represent the mean of four independent experiments (300 cells, ±SEM), two-sided Wilcoxon matched-pairs test, ****$p < 0.0001$. **c** Quantification and exemplary images of a PLA assay between Spike and RAB5A as in (**c**). Bars represent the mean of four independent experiments (400 cells, ±SEM), two-sided Wilcoxon matched-pairs test, ****$p < 0.0001$. DAPI (blue), nuclei. PLA signal (yellow). Scale bar, 20 μm. **d** Summary of the quantification shown in panels (**b**) (Spike-ACE2) and (**c**) (Spike-RAB5α) proximity upon SARS-CoV-2 infection. Individual data points and statistics are provided in panels (**b**) and (**c**). **e** Exemplary images of the colocalization (white) of PLA foci (red) indicating SARS-CoV-2 Spike and IFITM2 with early (EEA1, green, upper panel) or late endosomal markers (Rab7a, green, lower panel) in Calu-3 cells. Cells were infected with SARS-CoV-2 for 2 h at 4 °C or 2 h at 4 °C and 15 min at 37 °C as indicated. DAPI (blue), nuclei. **f** Quantification of the percentage of colocalization between the PLA signal and the indicated endosomal markers in 225 × 225 μm images. Bars represent the mean of four individual images (±SEM), unpaired $t$ test, **$p = 0.0031$. **g** Quantification of the combined PLA (IFITM2/Spike) in (**e**). Bars represent the mean of 120 cells (±SEM) over two independent experiments. The experiment was replicated twice to similar results. Two-sided Wilcoxon matched-pairs test, ****$p < 0.0001$. **a**–**c**, **e** Scale bars, 20 μm.

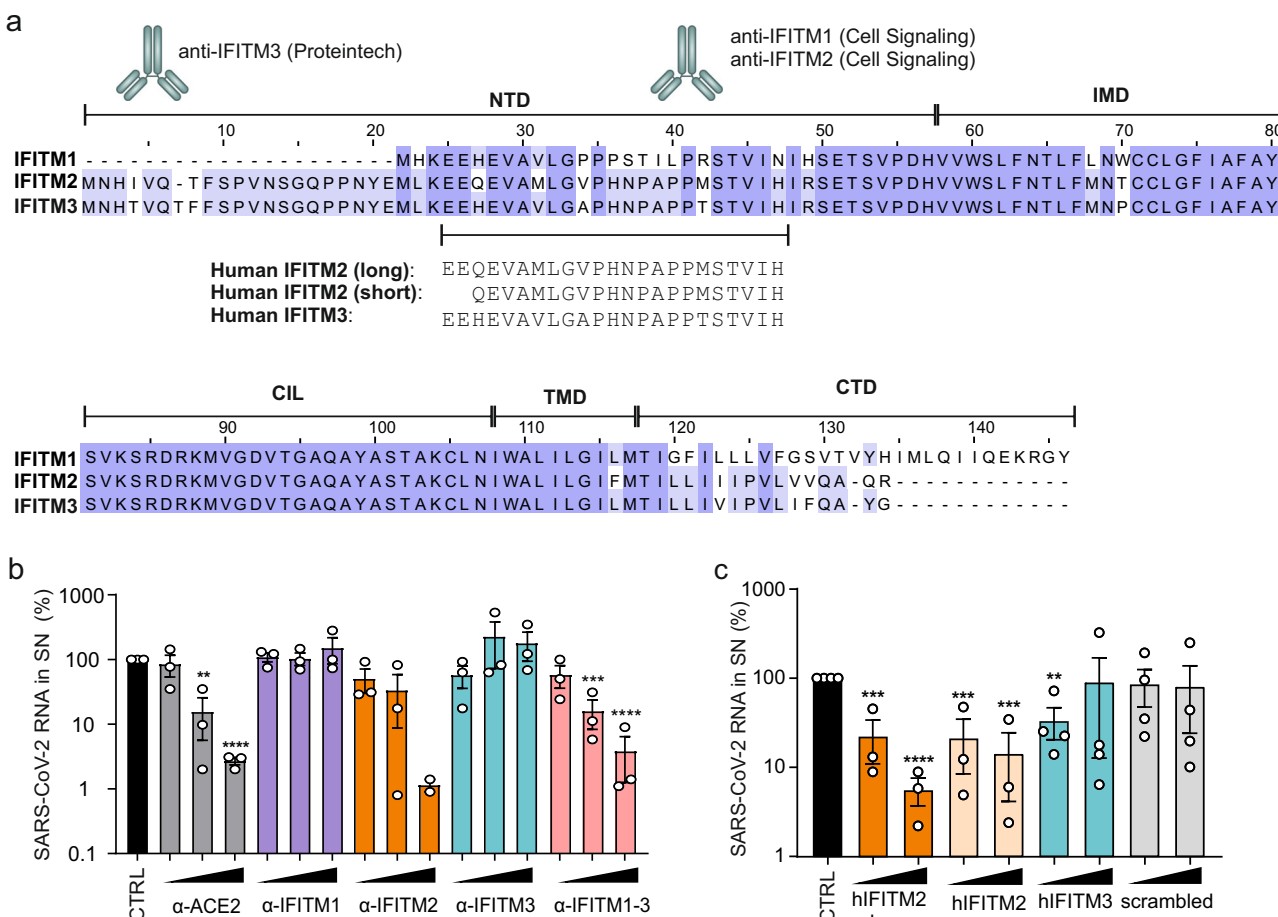

**Fig. 5 IFITM blocking antibodies and IFITM derived peptides target the N-terminal domain. a** Alignment of the amino acid sequence of human IFITM1, 2, and 3. Binding sites of IFITM blocking antibodies are indicated and the region of origin of the IFITM derived peptides highlighted. **b** Quantification of viral N RNA levels in the supernatant of Calu-3 cells treated with α-ACE2, α-IFITM1, α-IFITM2, α-IFITM3, and α-IFITM1-3 antibodies (7.5, 15, or 30 μg/ml) 1 h before infection (SARS-CoV-2, MOI 0.05), collected 48 h post-infection. Bars represent the mean of two (30 μg/ml α-IFITM2) or three (all other concentrations) independent experiments each measured in technical duplicates (±SEM), unpaired t-test. All p values were calculated compared to CTRL (ACE2, 15 μg/ml: 0.0011, ACE2, 30 μg/ml: <0.00001; IFITM2, 30 μg/ml: <0.00001; IFITM1-3, 15 μg/ml: 0.004; IFITM1-3, 30 μg/ml: <0.00001). **c** Quantification of viral N RNA levels in the supernatant of Calu-3 cells treated with IFITM-derived peptides (20 or 80 μg/ml) as indicated for 1 h before infection (MOI 0.05), collected 48 h post-infection. Bars represent the mean of three (hIFITM2 long and short peptide) or four (CTRL, hIFITM3 and scrambled) independent experiments each measured in technical duplicates (±SEM), unpaired $t$ test. All $p$ values were calculated compared to CTRL (hIFITM2 long, 20 μg/ml: 0.005; hIFITM2 long, 80 μg/ml: <0.0001; hIFITM2 short, 20 μg/ml: 0.0009; hIFITM2 short, 80 μg/ml: 0.0002; hIFITM3, 20 μg/ml: 0.0031). $p$ Values are indicated as *$p < 0.05$; **$p < 0.01$; ***$p < 0.001$; ****$p < 0.0001$ or were not significant ($p > 0.05$).

CoV-2 inhibiting IFITM2 antibody antibodies also efficiently impair SARS-CoV-2 replication (Fig. 5c). In contrast, a corresponding IFITM3-derived peptide, which differs in four of the 23 residues from the IFITM2-derived peptide, and a scrambled control peptide of the same length and amino acid composition had little if any effect on viral RNA yields. Notably, incubation of SARS-CoV-2 virions with the peptides prior to infection had no inhibitory effect (Supplementary Fig. 10). Thus, similarly to other inhibitors of SARS-CoV-2 infection[32,33], the IFITM2-derived peptides might target a region in the viral S protein that only becomes accessible during the entry process.

**IFITM-derived peptides or IFITM-targeting antibodies protect gut organoids and cardiomyocytes against SARS-CoV-2**. To better assess the potential relevance of IFITMs for viral spread and pathogenesis in SARS-CoV-2-infected individuals, we analyzed their expression in various cell types. We found that IFITM proteins are efficiently expressed in primary human lung bronchial epithelial (NHBE) cells, neuronal cells, and intestinal organoids derived from pluripotent stem cells (Supplementary Fig. 11a–c). These cell types and organoids represent the sites of SARS-CoV-2 entry and subsequent spread, i.e., the lung and the gastrointestinal tract[34–36], and the potential targets responsible for neurological manifestations of COVID-19[37]. Confocal microscopy analyses confirmed efficient induction of IFITM expression by IFN-β (Supplementary Fig. 12a). NHBE cells and cultures of neuronal cells did not support efficient SARS-CoV-2 replication precluding meaningful inhibition analyses. Gut organoids, however, are spermissive to SARS-CoV-2 replication[35], and treatment with the IFITM2-derived peptide or an antibody targeting the N-terminus of IFITMs strongly reduced viral RNA production (Supplementary Fig. 12b). Independent infection experiments confirmed that both agents significantly reduce viral N protein expression and cytopathic effects in gut organoids (Fig. 6a). Following up on recent evidence that SARS-CoV-2 causes cardiovascular disease[38], we investigated viral replication in human iPSC-derived cardiomyocytes. In agreement with published data[39], beating cardiomyocytes were highly susceptible to viral replication (Fig. 6b). All three IFITM proteins were expressed in cardiomyocytes and further induced by virus infection (Fig. 6c). On average, treatment of cardiomyocytes with the IFITM2- or IFITM3-derived peptides reduced the efficiency of SARS-CoV-2 replication by ~10- and 5-fold, respectively (Fig. 6d). In addition, treatment with these peptides suppressed or prevented the disruptive effects of virus infection on the ability of cardiomyocytes to beat in culture. Thus, IFITMs can be targeted to inhibit SARS-CoV-2 replication in cells from various human organs, including the lung, gut, and heart.

## Discussion

The present study demonstrates that endogenous expression of IFITMs is required for efficient replication of SARS-CoV-2 in human lung cells. In addition, we show that IFITMs can be targeted to inhibit SARS-CoV-2 infection of human lung, gut, and heart cells. These findings came as surprise since IFITMs have been reported to inhibit SARS-CoV, MERS-CoV, and, very recently, SARS-CoV-2 S-mediated infection[10,12,18,19]. Confirming and expanding these previous studies, we show that artificial overexpression of IFITM proteins in HEK293T cells prevents S-mediated VSVpp and HIVpp fusion as well as genuine SARS-CoV-2 entry. However, exactly the opposite was observed for genuine SARS-CoV-2 upon manipulation of endogenous IFITM expression in human lung cells: silencing of all three IFITM proteins reduced SARS-CoV-2 infection, although IFITM2 typically showed the strongest effects. Lack of IFITM expression

strongly reduced viral RNA yields but had even stronger effects on the infectious virus titers suggesting that IFITM might affect both susceptibilities to infection as well as the infectiousness of progeny SARS-CoV-2 virions. Our results provide unexpected insights into the role of IFITM proteins in the spread and pathogenesis of SARS-CoV-2 and suggest that these supposedly antiviral factors are hijacked by SARS-CoV-2 as cofactors for efficient entry.

While wildtype IFITM proteins have generally been described as inhibitors of SARS and MERS coronaviruses specific point mutations may convert IFITM3 from an inhibitor to an enhancer Spike-mediated pseudoparticle transduction[10,40]. It has been reported that overexpression of IFITM3 promotes infection by hCoV-OC43, one of the causative agents of common colds[26]. However, IFITM3 seemed to be least relevant for SARS-CoV-2 infection in the present study. Thus, although both human coronaviruses may hijack IFITMs for efficient infection they show distinct preferences for specific IFITM proteins. It is under debate whether SARS-CoV-2 mainly fuses at the cell surface or in endosomes and cell-type-specific differences may explain why IFITM2 plays a key role in Calu-3 cells, while IFITM1 is at least as important in SAEC cells. Most importantly, our results clearly demonstrate that IFITM proteins act as critical cofactors for efficient SARS-CoV-2 infection in in vitro conditions closest to the in vivo situation.

We currently do not yet fully understand why overexpressed and endogenous IFITM proteins have opposite effects on SARS-CoV-2 infection. However, artificial overexpression may change the topology, localization and endocytic activity of proteins. The exact topology of IFITMs is unknown[8] and it has been reported that specific mutations in IFITM3 affecting these features may convert IFITM3 from an inhibitor to an enhancer of coronavirus infection[10,40]. The antiviral activity of IFITMs is very broad and does usually not involve interactions with specific viral glycoproteins[7,8], although it has been reported that the HIV-1 Envelope protein interacts with IFITMs[41]. In contrast, different lines of experimental evidence suggest that the ability of SARS-CoV-2 to hijack IFITMs for efficient entry involves specific interactions between the N-terminal region of IFITMs and the viral S protein (outlined in Supplementary Fig. 13). Altogether, our results support that a surface accessible N-terminal region of IFITM2 interacts with the S protein and promotes SARS-CoV-2 fusion predominantly in early endosomes.

IFITMs are strongly induced during the innate immune response in SARS-CoV-2-infected individuals[42,43]. Thus, utilization of IFITMs as infection cofactors may promote SARS-CoV-2 invasion of the lower respiratory tract as well as spread to secondary organs, especially under inflammatory conditions. Further studies are required but the efficient expression in neurons and cardiomyocytes suggests that IFITMs may play a role in the well-documented neuronal and cardiovascular complications associated with SARS-CoV-2 infection[44,45]. Perhaps most intriguingly, we show that IFITM-derived peptides and antibodies raised against the N-terminal region of IFITM2 efficiently inhibit SARS-CoV-2 replication. Notably, the inhibitory antibodies need to target a native epitope on IFITM2 accessible from the cell surface. While IFITM proteins are mainly known for their antiviral activity they were also reported to be involved in physiological functions[15,46], such as amplification of PI3K signaling in B cells[47]. Although targeting cellular IFITM proteins as a therapeutic approach may come at a cost, it should reduce the risk of viral resistance and warrants further investigation.

## Methods

**Cell culture**. All cells were cultured at 37 °C in a 5% $CO_2$ atmosphere. Human embryonic kidney 293T cells (HEK293T; ATCC) were maintained in Dulbecco's Modified Eagle Medium (DMEM) supplemented with 10% heat-inactivated fetal

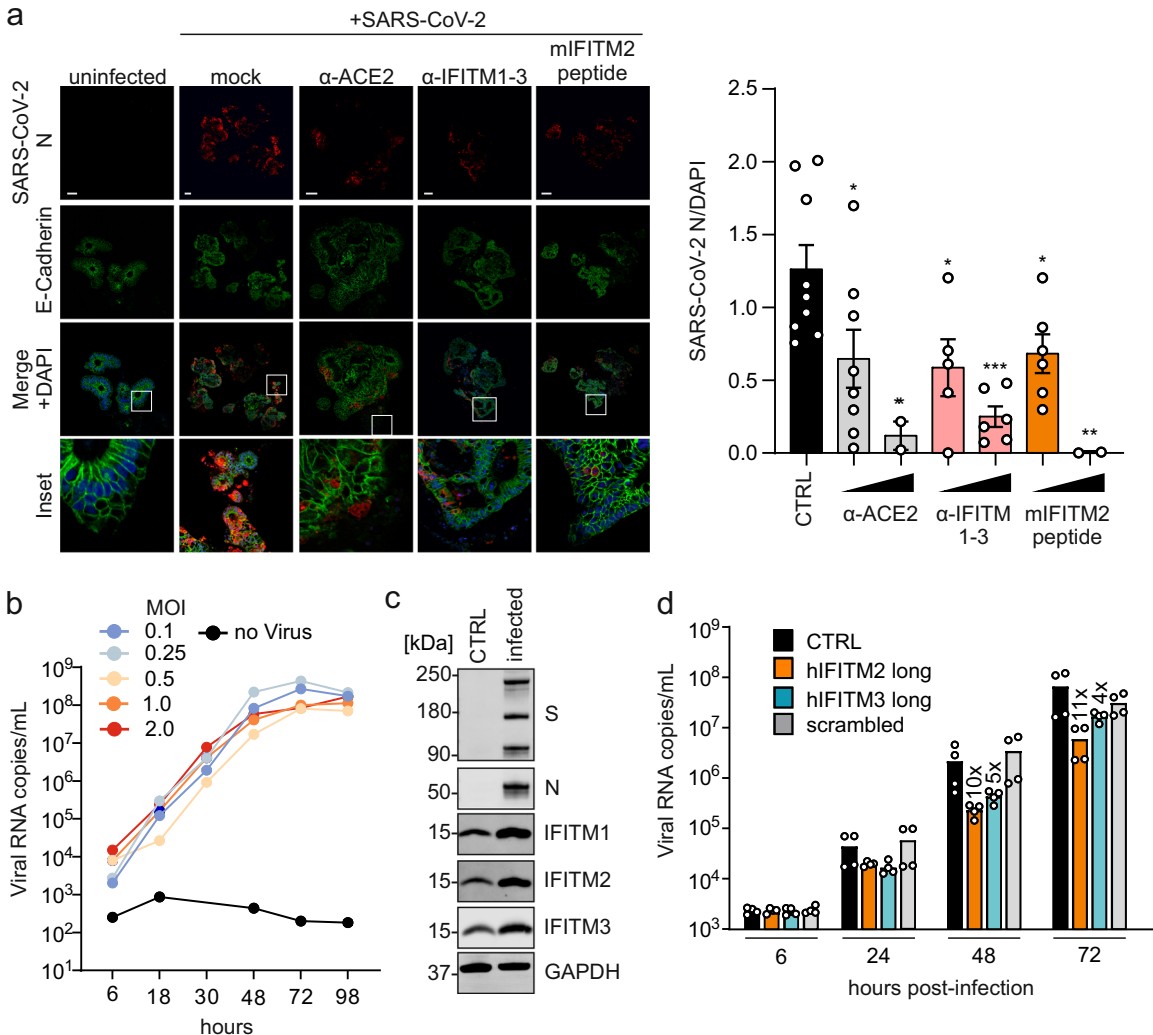

**Fig. 6 Blocking antibodies and IFITM-derived peptides treatment decrease SARS-CoV-2 infection in gut organoids and cardiomyocytes. a**
Immunohistochemistry of gut organoids treated with blocking antibodies and a mouse IFITM2 (mIFITM2) derived peptide and infected with SARS-CoV-2 (MOI 0.15). Organoids were stained with α-SARS-CoV-2 N (red), E-Cadherin (green), and DAPI (blue). Scale bar, 100 μm (left panel). Quantification of SARS-CoV-2 N fluorescence, normalized to DAPI (right panel). Bars represent the mean of CTRL:9, α-ACE2 15 μg/ml:8, α-ACE2 30 μg/ml:2, α-IFITM1-3 15 μg/ml:5, α-IFITM1-3 30 μg/ml:6, mIFITM2 peptide 15 μg/ml:6, or mIFITM2 peptide 30 μg/ml:2 individual images over one experiment (±SEM), unpaired t-test with Welch's correction All $p$ values were calculated compared to CTRL (α-ACE2 15 μg/ml: 0.0104, α-ACE2 30 μg/ml: 0.0093, α-IFITM1-3 30 μg/ml: 0.0002, mIFITM2 peptide 15 μg/ml: 0.0295, mIFITM2 peptide 30 μg/ml: 0.005). **b** Quantification of viral N RNA levels in the supernatant of SARS-CoV-2 infected cardiomyocytes (increasing MOIs as indicated) at indicated time points. Lines represent a single experiment measured in duplicates. **c** Immunoblot of IFITM1, IFITM2, and IFITM3 in cardiomyocytes infected with SARS-CoV-2. Whole-cell lysates were stained with α-IFITM1, α-IFITM2, α-IFITM3, and α-GAPDH. $n=1$. **d** Quantification of viral N RNA levels in the supernatant of SARS-CoV-2 infected cardiomyocytes (MOI 0.1) treated with IFITM-derived peptides, collected at indicated time points post-infection. Bars represent the mean of two independent experiments each measured in technical duplicates. Non-infected controls were below the quantification level (<1000). Exact $p$ values are provided in Supplementary Data 1. $p$ Values are indicated as *$p < 0.05$; **$p < 0.01$; ***$p < 0.001$; ****$p < 0.0001$ or were not significant ($p > 0.05$).

calf serum (FCS), L-glutamine (2 mM), streptomycin (100 μg/ml), and penicillin (100 U/ml). HEK293T was provided and authenticated by the ATCC while Hela-ACE2 was generated using the ACE2 expression construct. Caco-2 (human epithelial colorectal adenocarcinoma) cells were maintained in DMEM containing 10% FCS, glutamine (2 mM), streptomycin (100 μg/ml) and penicillin (100 U/ml), NEAA supplement (Non-essential amino acids (1 mM)), sodium pyruvate (1 mM). Calu-3 (human epithelial lung adenocarcinoma) cells were cultured in Minimum Essential Medium Eagle (MEM) supplemented with 10% FCS (during viral infection) or 20% (during all other times), penicillin (100 U/ml), streptomycin (100 μg/ml), sodium pyruvate (1 mM), and NEAA supplement (1 mM). Hybridoma cells (Mouse I1 Hybridoma CRL-2700; ATCC) were cultured in Roswell Park Memorial Institute (RPMI) 1640 medium supplemented with 10% FCS, L-glutamine (2 mM), streptomycin (100 μg/ml), and penicillin (100 U/ml). Vero cells (ATCC, CCL-81) cells were maintained in DMEM containing 2.5% FCS, glutamine (2 mM), streptomycin (100 μg/ml), and penicillin (100 U/ml), NEAA supplement (Non-essential amino acids (1 mM)), sodium pyruvate (1 mM). Monoclonal anti-

VSV-G containing supernatant was aliquoted and stored at −20 °C. NHBE (primary human bronchial/tracheal epithelial, Lonza) cells were grown in Bronchial Epithelial Cell Growth Basal Medium (BEGM, Lonza) and Bronchial Epithelial Cell Growth Medium SingleQuots Supplements and Growth Factors (Lonza). SAEC, Lonza were grown in SAEC Growth Basal Medium (SABM, Lonza) and SAEC Growth Medium SingleQuots Supplements and Growth Factors (Lonza).

**Human hESC cultivation and gut organoids differentiation.** Human embryonic stem cell (hESC) line HUES8 (Harvard University) was used with permission from the Robert Koch Institute according to the "Approval according to the stem cell law" AZ 3.04.02/0084. Cells were cultured on hESC Matrigel (Corning) in mTeSR1 medium (Stemcell Technologies) at 5% $CO_2$ and 37 °C. The medium was changed every day and cells were split twice a week with TrypLE Express (Invitrogen). Experiments involving human stem cells were approved by the Robert-Koch Institute (Approval according to the stem cell law 29.04.2020).

**Cardiomyocyte differentiation**. Human episomal hiPSCs (#A18945, Thermo Fisher Scientific) at passage 2 were split using TrypLE (#12604-013, Thermo Fisher Scientific) to generate a single-cell suspension. 18000 iPS cells were seeded on Geltrex (#A1413302, Thermo Fisher Scientific) matrix coated 12 well plates. 3 days post-splitting differentiation protocol into iPS cardiomyocytes using the PSC cardiomyocytes Differentiation Kit (#A29212-01, Thermo) was initiated. Contracting iPSC-derived cardiomyocytes were present 14 days post-differentiation initiation.

**Neuronal differentiation**. Human iPSC, either generated from keratinocytes as previously described[48] or commercially purchased from the iPSC Core facility of Cedars Sinai (Los Angeles, California), were cultured at 37 °C (5% CO₂, 5% O₂) on Matrigel-coated (Corning, 354277) 6-well plates using mTeSR1 medium (Stem Cell Technologies, 83850). Generation was performed in agreement with the ethical committee of the Ulm University (approval Nr.0148/2009 and 265/12) and in compliance with the guidelines of the Federal Government of Germany. The use of human material was approved by the Declaration of Helsinki concerning Ethical Principles for Medical Research Involving Human Subjects. All participants gave informed consent for the study. Neuronal differentiation was chemically induced by culturing hiPSC (hiPSCs were obtained from 3 donors without known pathologies. Control 1 is a 45-year-old female; Control Biocat is a 65-year-old male, and Control 4 is 49 years old man) colonies in suspension in ultra-low attachment T75 flasks (Corning, 3815), to allow the formation of embryody bodies (EBs). During the first 3 days of differentiation, cells were cultivated in DMEM/F12 (Gibco, 31331-028) containing 20% knockout serum replacement (Gibco, 10828028), 1% NEAA, 1% β-mercaptoethanol, 1% antibiotic-antimycotic, SB-431542 10 µM (Stemcell Technologies, 72232), Dorsomorphin 1 µM (Tocris, 3093), CHIR 99021 3 µM (Stemcell Technologies, 72054), Pumorphamine 1 µM (Miltenyi Biotec, 130-104-465), Ascorbic Acid 200 ng/µL, cAMP 500 µM (Sigma-Aldrich, D0260), 1% supplement (Stemcell Technologies, 05731), 0.5% N2 supplement (Gibco, 17502-284). From the fourth day on, the medium was switched to DMEM/F12 added with 24 nM sodium selenite (Sigma-Aldrich, S5261), 16 nM progesterone (Sigma-Aldrich, P8783), 0.08 mg/mL apo transferrin (Sigma-Aldrich, T2036), 0.02 mg/mL, Insulin (Sigma-Aldrich, 91077 C), 7.72 µg/mL putrescine (Sigma-Aldrich, P7505), 1%NEAA, 1% antibiotic-antimycotic, 50 mg/mL heparin (Sigma-Aldrich, H4783), 10 µg/mL of the neurotrophic factors BDNF (Peprotech, 450-02), GDNF (Peprotech, 450-10), and IGF1 (Peprotech, 100-11), 10 µM SB-431542, 1 µM dorsomorphin, 3 µM CHIR 99021, 1 µM pumorphamine, 150 µM. vitamin C, 1 µM retinoic acid, 500 µM cAMP, 1% Neurocult supplement, 0.5% N2 supplement. After 5 further days, neurons were dissociated to single-cell suspension and plated onto µDishes, or 6-well plates (Corning, 353046) pre-coated with GFR Matrigel (Corning, 356231).

**Expression constructs**. Expression plasmids encoding for IFITM1, IFITM2 and IFITM3 (pCG_IFITM1, pCG_IFITM2, pCG_IFITM3 and pCG_IFITM1-IRES_eGFP, pCG_IFITM2-IRES_eGFP and pCG_IFITM3-IRES_BFP) were PCR amplified and subcloned in pCG based backbones using flanking restriction sites XbaI and MluI. The IFITM2 Y19D mutant was kindly provided by Christine Goffinet. pCG_SARS-CoV-2-Spike-IRES_eGFP (humanized), encoding the spike protein of SARS-CoV-2 isolate Wuhan-Hu-1, NCBI reference Sequence YP_009724390.1 while pCG_SARS-CoV-2-Spike C-V5-IRES_eGFP was PCR amplified and subcloned using XbaI+MluI, while pCG_SARS-CoV2-Spike C-V5-IRES_eGFP was PCR amplified and subcloned using XbaI+MluI. To generate the pLV-EF1a-human ACE2-IRES-puro, pTargeT-hACE2 were provided by Sota Fukushi and Masayuki Saijo (National Institute of Infectious Diseases, Tokyo, Japan). The ORF of ACE2 was extracted with MluI and SmaI and then inserted into the MluI-HpaI site of pLV-EF1a-IRES-Puro.

**Pseudoparticle stock production**. To produce pseudotyped VSV(luc/GFP)ΔG particles, HEK293T cells were transfected with pCG_SARS-CoV-2-Spike C-V5-IRES_GFP using polyethyleneimine (PEI)[49]. Twenty-four hours post-transfection, the cells were infected with VSVΔG(GFP/luc)*VSV-G at an MOI of 1. The inoculum was removed after 1 h. Pseudotyped particles were harvested at 16 h post-infection. Cell debris was removed by centrifugation at 805 × g for 5 min. Residual input particles carrying VSV-G were blocked by adding 10% (v/v) of I1 Hybridoma supernatant (I1, mouse hybridoma supernatant from CRL-2700; ATCC) to the cell culture supernatant. To produce pseudotyped HIV-1(fLuc)Δenv particles, HEK293T cells were transfected with pCMVdR8.91 (Addgene) and pSEW-luc2 (Promega, # 9PIE665) or pCMV4-BlaM-vpr (Addgene, #21950) as well as pCG_SARS-CoV-2-Spike C-V5-IRES_eGFP using TransIT-LT1 according to the manufacturer's protocol. Six hours post-transfection, the medium was replaced with DMEM containing only 2.5% FCS. The particles were harvested 48 h post-transfection. Cell debris was pelleted by centrifugation at 805 × g for 5 min.

**Target cell assay**. HEK293T cells were transiently transfected using PEI[49] with pLV-EF1a-human ACE2-IRES-puro and pCG-IFITM1-IRES_eGFP or pCG-IFITM2-IRES_eGFP or pCG-IFITM3-IRES_BFP. 24 h post-transfection, cells were transduced/infected with HIV-1Δenv(fLuc)* SARS-CoV-2 S or VSV(luc) ΔG*SARS-CoV-2 S particles. Sixteen hours post-infection Luciferase activity was quantified.

**Producer cell assay**. HEK293T cells were transiently transfected using PEI with pCG_IFITM1 IRES_eGFP, pCG_IFITM2-IRES_eGFP, pCG_IFITM3-IRES_BFP or pCG_SARS-CoV-2-Spike-C-V5-IRES_eGFP. Twenty-four-hour post-transfection cells were infected with VSV(GFP)ΔG*SARS-CoV-2 S or VSV(GFP)ΔG*VSV-G with an MOI of 1. After 1 hour, the inoculum was removed and the medium was replaced. Sixteen hours post-infection, cells were harvested to generate cell lysates. The supernatant was collected separately and supplemented with 10 % (v/v) of I1 Hybridoma Supernatant. Blocked supernatants were used to infect Caco-2 cells. 16 h post-infection medium was removed and cells were washed with PBS, detached with Trypsin-EDTA (Gibco), and fixed with 2% PFA for 30 min at 4 °C. GFP-positive cells were analyzed by flow cytometry (BD Canto II, data acquisition using BD FACS Diva software, and FlowJo).

**Luciferase assay**. To determine viral gene expression, the cells were lysed in 300 µl of Luciferase Lysis buffer (Luciferase Cell Culture Lysis, Promega), and firefly luciferase activity was determined using the Luciferase Assay Kit (Luciferase Cell Culture, Promega) according to the manufacturer's instructions on an Orion microplate luminometer (Berthold).

**Vpr-BlaM fusion assay**. HEK293T cells were seeded and transiently transfected using PEI[49] with pLV-EF1a-human_ACE2-IRES-puro and pCG_IFITM1, pCG_IFITM2, or pCG_IFITM3. Twenty-four-hour post-transfection, cells were transferred to a 96-well plate. On the next day, cells were infected with 50 µl HIV-1Δenv(BlaM-Vpr)-*SARS-CoV-2-S particles for 2.5 h at 37 °C, followed by washing with PBS. Cells were detached and stained with CCF2/AM (1 mM)[50]. Finally, cells were washed and fixed with 4% PFA. The change in emission fluorescence of CCF2 after cleavage by the BlaM-Vpr chimera was monitored by flow cytometry using a FACSCanto II (BD) and BD FACS Diva and FlowJo for analysis.

**SARS-CoV-2 virus stock production**. BetaCoV/Netherlands/01/NL/2020 or BetaCoV/ France/IDF0372/2020 was propagated on Vero E6 infected at an MOI of 0.003 in serum-free medium containing 1 µg/ml trypsin[24]. In brief, the cells were inoculated for 2 h at 37 °C before the inoculum was removed. The supernatant was harvested 48 h post-infection upon visible cytopathic effect (CPE). To remove the debris, the supernatants were centrifuged for 5 min at 1000 × g, then aliquoted and stored at −80 °C. Infectious virus titer was determined as plaque-forming units (PFU).

**PFU assay**. The PFU assay was performed according to a published protocol[24]. SARS-CoV-2 stocks were serially diluted and confluent monolayers of Vero E6 cells infected. After incubation for 2 h at 37 °C with shaking every 20 min. The cells were overlaid with 1.5 ml of 0.8% Avicel RC-581 (FMC) in medium and incubated for 3 days. Cells were fixed with 4% PFA at room temperature for 45 min. After the cells were washed with PBS once 0.5 ml of staining solution (0.5% crystal violet and 0.1% triton in water). After 20 min incubation at room temperature, the staining solution was removed using water, virus-induced plaque formation quantified, and PFU per ml calculated.

**TCID50 endpoint titration**. SARS-CoV-2 stocks or infectious supernatants were serially diluted. Totally, 25,000 Caco-2 cells were seeded per well in 96 flat bottom well plates in 100 µl medium and incubated overnight. Next, 50 µl of titrated SARS-CoV-2 stocks or supernatants were used for infection, resulting in final dilutions of 1:101 to 1:1012 on the cells in triplicates. Cells were then incubated for 5 days and monitored for CPE. TCID50/ml was calculated according to the Reed and Muench method.

**qRT-PCR**. N (nucleoprotein) RNA levels were determined in supernatants of cells collected from SARS-CoV-2 infected cells 6, 24, or 48 h post-infection. Total RNA was isolated using the Viral RNA Mini Kit (Qiagen) according to the manufacturer's instructions. qRT-PCR was performed according to the manufacturer's instructions using TaqMan Fast Virus 1-Step Master Mix (Thermo Fisher) and an OneStepPlus Real-Time PCR System (96-well format, fast mode). Primers were purchased from Biomers and dissolved in RNAse free water. Synthetic SARS-CoV-2-RNA (Twist Bioscience) were used as a quantitative standard to obtain viral copy numbers. All reactions were run in duplicates using TaqMan primers/probes (Supplementary Table 1 HKU-NF, -NR and -NP; Thermo Fisher).

**IFITM1, 2, and 3 KD**. Twenty-four and 96 h after seeding, Calu-3 or SAEC cells were transfected twice with 20 µM of either non-targeting siRNA or IFITM1, IFITM2, or IFITM3 specific siRNA using Lipofectamine RNAiMAX (Thermo Fisher) according to the manufacturer's instructions. 14 h post-transfection, the medium was replaced with fresh medium supplemented with 500 U/ml IFN-β in the indicated conditions. For SAEC experiments cells were stimulated with 40 U/ml IFN-γ or 500U/ml IFN-β in the indicated conditions. Seven hour after the second transfection, Calu-3 or SAEC cells were infected with SARS-CoV-2 with an MOI of 0.05 and 2.5, respectively. Six-hour later, the inoculum was removed, cells were washed once with PBS and supplemented with fresh media. Forty-eight hours post-infection, cells and supernatants were harvested for Western blot and qRT-PCR

analysis respectively. IFITMs mRNA levels were determined using specific customized primers and probes (Supplementary Table 1; IDT).

**Stimulation with type I and II interferons.** Calu-3, NHBE cells, and SAEC cells were seeded in 12-well plates. For the gut organoids stimulation, HUES88 were seeded in 24-well-plates were coated with growth factor reduced (GFR) Matrigel (Corning) and in mTeSR1 with 10 µM Y-27632 (Stemcell technologies). The next day, differentiation to organoids was started at 80-90% confluency. Cells or organoids were stimulated with IFN-α (500 U/ml, R&D systems 11100-1), IFN-β (500 U/ml, R&D systems 8499-IF-010) or IFN-γ (200 U/ml, R&D systems 285-IF-100). Three days post-stimulation whole cell lysates were generated.

**Cardiomyocytes infection and kinetics.** Human iPSC-derived cardiomyocytes were cultures in 12-wells plates until they were 3–4 weeks old and homogenously beating. Cells were infected with increasing MOIs (0.1, 0.25, 0.5, 1, and 2) of the BetaCoV/Netherlands/01/NL/2020 strain. Six-hour post-infection, cells were washed once with PBS to remove input virus and supplemented with fresh media. Virus-containing supernatant was harvested every day and replaced with fresh media until day 7 (as indicated). N gene RNA copies were determined by qRT-PCR and cells were harvested for Western blot analysis at the latest time point.

**Peptides synthesis.** The IFITM-derived peptides were synthesized by UPEP, Ulm using F-moc chemistry. Purification to homogeneity of more than 95% was done by reverse-phase HPLC. Peptide stock was prepared in distilled water to a final concentration of 10 mg/ml.

**Inhibition by IFITM antibodies and peptides.** Calu-3 cells were seeded in 48-well format (peptides assays), or in 24-well format (antibodies assay), 24 h later cells were treated with increasing concentrations (20 and 80 µg/ml) of IFITMs derived peptides (human IFITM2 long: EEQEVAMLGVPHNPAPPMSTVIH, human IFITM2 short: QEVAMLGVPHNPAPPMST-VIH, mouse IFITM2 (mIFITM2) long: EEYGVTELGEPSNSAVVRTTVIN, human IFITM3 long: EEHEVAVLGAPHNPAPPTSTVIH, scrambled IFITM2: EGESGVTTATVEVVIER-NN-LPY) or blocking antibodies (15 and 30 µg/ml) (α-ACE2 AK (AC18Z), Santa Cruz Biotechnology sc-73668; α-IFITM1 Cell Signaling 13126 S, α-IFITM2 Cell Signaling 13530 S, α-IFITM3 Proteintech 11714-1-AP, α-IFITM1/2/3 (F-12) Santa Cruz Biotechnology sc-374026) as indicated. 2 h post-treatment, cells were infected with SARS-CoV-2 with an MOI of 0.05. 6 h post-infection, cells were washed once with PBS and supplemented with fresh MEM medium. 48 h post-infection supernatants were harvested for qRT-PCR analysis. Cardiomyocytes were seeded in 12-well plates and treated with 100 µg/ml of indicated peptides 1 h prior to infection (MOI 0.01). 6 h post-infection, cells were washed once with PBS to remove input virus and supplemented with fresh media. Virus-containing supernatant was harvested every day, replaced with fresh media until day 3, and fresh peptides (100 µg/ml) (as indicated). N gene RNA copies were determined by qRT-PCR. Gut organoids were treated with increasing concentrations (15 and 30 µg/ml) of IFITMs derived peptides (mouse IFITM2 (mIFITM2) antibody blocking peptide Santa Cruz sc-373676 P) and blocking antibodies (α-ACE2 AK (AC18Z), Santa Cruz Biotechnology sc-73668, α-IFITM1/2/3 (F-12) Santa Cruz Biotechnology sc-374026) as indicated. One hour thirty minutes post-treatment, organoids were infected with SARS-CoV-2 with an MOI of 0.15[51]. Forty-eight hours post-infection gut organoids were harvested for qRT-PCR analysis.

**Virus treatment.** Calu-3 cells were seeded in 48-wells, 24 h later SARS-COV-2 (0.05 MOI) was incubated for 30 min at 37˚C with indicated concentrations of IFITM-derived peptides. Fifty microlitres of the inoculum were used to infect the cells. Six-hour later cells were supplemented with fresh medium. Forty-eight hours post-infection supernatants were harvested for qRT-PCR analysis.

**Immunofluorescence of gut organoids.** For histological examination, organoids were fixed in 4 % PFA overnight at 4 °C, washed with PBS, and pre-embedded in 2% agarose (Sigma) in PBS. After serial dehydration, intestinal organoids were embedded in paraffin, sectioned at 4 µm, deparaffinized, rehydrated, and subjected to heat-mediated antigen retrieval in tris Buffer (pH 9) or citrate buffer (pH 6). Sections were permeabilized with 0.5% Triton-X for 30 min at RT and stained overnight with primary antibodies (rabbit anti-IFITM1 Cell Signaling 13126S, 1:500 or rabbit anti-IFITM2 Cell Signaling #13530S, 1:500 or rabbit anti-IFITM3 Cell Signaling #59212S, 1:250 or anti-SARS-CoV-2 N 1:500 or anti-E-Cadherin 1:500) diluted in antibody diluent (Zytomed) in a wet chamber at 4 °C. After washing with PBS-Tween 20, slides were incubated with secondary antibodies (Alexa Fluor IgG H + L, Invitrogen, 1:500) and 500 ng/ml DAPI in Antibody Diluent for 90 min in a wet chamber at RT. After washing with PBS-T and water, slides were mounted with Fluoromount-G (Southern Biotech). Negative controls were performed using IgG controls or irrelevant polyclonal serum for polyclonal antibodies, respectively. Cell borders were visualized by E-cadherin staining. Images were acquired using an LSM 710 system.

**GFP split fusion assay.** GFP1-10 and GFP11-expressing HEK293T cells were seeded separately in a 24-well plate. One day post-seeding, cells were transiently transfected using the calcium-phosphate precipitation method[52]. GFP1-10 cells were co-transfected with increasing amounts (0, 8, 16, 32, 64, 125, 250, 500 ng) of pCG_IFITM1, pCG_IFITM2, pCG_IFITM3, and 250 ng of pLV-EF1a-human ACE2-IRES-puro. GFP11 cells were transfected with 250 ng of pCG_SARS-CoV-2-Spike C-V5 codon-optimized. Sixteen hours post-transfection, GFP1-10, and GFP11 cells were co-cultured in a poly-L-lysine-coated 24-well plate. Twenty-four-hour post-coculturing, cells were fixed with 4% PFA and cell nuclei were stained using NucRed Live 647 ReadyProbes Reagent (Invitrogen) according to the manufacturer's instructions. Fluorescence imaging of GFP and NucRed was performed using a Cytation3 imaging reader (BioTek Instruments). 12 images per well were recorded automatically using the NucRed signal for autofocusing. The GFP area was quantified using ImageJ.

**Whole-cell lysates.** To determine the expression of cellular and viral proteins, cells were washed in PBS and subsequently lysed in Western blot lysis buffer (150 mM NaCl, 50 mM HEPES, 5 mM EDTA, 0.1% NP40, 500 µM Na₃VO₄, 500 µM NaF, pH 7.5) supplemented with protease inhibitor (1:500, Roche)[49]. After 5 min of incubation on ice, samples were centrifuged (4 °C, 20 min, 20,817 × g) to remove cell debris. The supernatant was transferred to a fresh tube, the protein concentration was measured and adjusted using Western blot lysis buffer. Lysates from iPSC-derived neurons were prepared following previously published protocols[53]. Briefly, neurons were harvested in cold PBS (Gibco) and centrifuged at 2655 × g for 3 min. Pellets were then resuspended and incubated at 4 °C on an orbital shaker for 2 h in RIPA buffer. The lysate was then sonicated and protein concentration was determined by Bradford assay.

**Sodium dodecyl sulfate-polyacrylamide gel electrophoresis and immunoblotting.** Western blotting was performed according to a recently published protocol[49]. In brief, whole-cell lysates were mixed with 4× or 6× Protein Sample Loading Buffer (LI-COR, at a final dilution of 1×) supplemented with 10% β-mercaptoethanol (Sigma Aldrich), heated at 95 °C for 5 min, separated on NuPAGE 4 ± 12% Bis–Tris Gels (Invitrogen) for 90 min at 100 V and blotted onto Immobilon-FL PVDF membranes (Merck Millipore). The transfer was performed at a constant voltage of 30 V for 30 min. After the transfer, the membrane was blocked in 1% Casein in PBS (Thermo Scientific). Proteins were stained using primary antibodies against IFITM1 (α-IFITM1, Cell Signaling #13126S, 1:1000,), IFITM2 (α-IFITM2 Cell Signaling #13530 S, 1:1000), IFITM3 (α-IFITM3 Cell Signaling #59212S, 1:1000) SARS Spike CoV-2 (SARS-CoV-1/-2 (COVID-19) spike antibody [1A9], GTX-GTX632604, 1:1000), VSV-M (Mouse Monoclonal Anti-VSV-M Absolute antibody, ABAAb01404-21.0, 1:1000), actin (Anti-beta Actin antibody Abcam, ab8227, 1:5000,), ACE2 (Rabbit polyclonal anti-ACE2 Abcam, ab166755, 1:1000); rabbit anti-V5 (Cell Signaling #13202; 1:1000), mouse anti-FLAG (Sigma #F1804; 1000), rat anti-GAPDH (Biolegend #607902, 1:1000) and SARS CoV-2N (anti-SARS-CoV-2N Sino Biologicals #40588-V08B, 1:1000).and Infrared Dye labeled secondary antibodies (LI-COR IRDye). Membranes were scanned using LI-COR and band intensities were quantified using Image Studio (LI-COR).

**Proximity ligation assay.** Calu-3 or SAEC were seeded in a 24-well plate on a coverslip glass. Twenty-four hour and 72 h post-seeding, the cells were transfected with 20 µM either non-targeting siRNA or IFITM1 or IFITM3 siRNAs using RNAimax according to the manufacturer's instructions. For the PLA in Hela-ACE2 cells, the latter were seeded on 24-well glass slides and transfected with either 0.5 µg IFITM2 wt or IFITM2 Y19D expression constructs. Prior to infection, cells were pre-chilled for 30 min at 4 °C and then infected with VSV(luc)ΔG*-SARS-CoV-2 S (MOI 2) or BetaCoV/France/IDF0372/2020 (MOI 1.5) for 2 h on ice. Cells have been washed once with cold PBS and fixed with 4% PFA. For staining following antibodies were used: IFITM1 (α-IFITM1 Cell Signaling 13126S), IFITM2 (α-IFITM2 Abcam 236735), IFITM3 (α-IFITM3 Cell Signaling 59212S), SARS Spike CoV-2 (SARS-CoV / SARS-CoV-2 (COVID-19) spike antibody [1A9], GTX-GTX632604), Rab5 alpha (Rab5 (RAB5A) Goat Polyclonal Antibody Origene AB0009-200) and ACE2 (Rabbit polyclonal anti-ACE2 Abcam, ab166755). All in a concentration 1:100. Images were acquired on a Zeiss LSM 710 and processed using ImageJ (Fiji)[54].

**Co-immunoprecipitation SARS-CoV-2 Spike and IFITMs.** HEK293Ts were transfected using PEI with 0.5 µg pCG-SARS CoV2 Spike-V5 and 0.5 µg of pCG IFITM1, IFITM2, or IFITM3. 24 h later, samples were lysed with IP lysis buffer (50 mM, Tris pH8, 150 mM NaCl, 1 % NP40, protease inhibitor) for 10 min on ice. Lysed samples were centrifuged and incubated for 3 h with Pierce Protein A/G Magnetic beads (88802) which were pre-incubated overnight with V5 antibody (Cell signaling E9H80; 5 µg of primary antibody per 10 µl of beads per sample).

**MaMTH assay**[31,55]. Human IFITM proteins and SARS-CoV-2 viral proteins were cloned into MaMTH N-term tagged Prey and C-term tagged Bait vectors respectively using Gateway cloning technology (ThermoFisher). The correctness of recombined insertions was confirmed by Sanger sequencing (Eurofins). HEK293T

B0166 Gaussia luciferase reporter cells were co-transfected in 96-well plates with 25 ng SARS-CoV-2 protein Bait and 25 ng IFITM or control protein Prey MaMTH vectors in triplicates using PEI transfection reagent. Gal4 (transcription factor), as well as EGFR Bait with SHC1 Prey, served as positive controls, whereas SARS-CoV-2 Bait proteins with Pex7 Prey were used as negative controls. The following day, Bait protein expression was induced with 0.1 μg/ml doxycycline. Cell-free supernatants were harvested 2 days post-transfection and the released Gaussia reporter was measured 1 s after injecting 20 mM coelenterazine substrate using an Orion microplate luminometer. To determine the level of protein interaction, Gaussia values were normalized to Pex7 Prey negative control for each Bait. Bait and Prey protein expression levels were determined by Western blotting.

**Statistics and reproducibility**. Statistical analyses were performed using Graph-Pad PRISM 8 (GraphPad Software). $p$ Values were determined using a two-tailed Student's $t$ test with or without Welch's correction or two-sided Wilcoxon matched-pairs test. Unless otherwise stated, data are shown as the mean of at least three independent experiments ± SEM. Significant differences are indicated as $*p < 0.05$; $**p < 0.01$; $***p < 0.001$. Statistical parameters are specified in the figure legends and all $p$ values are listed in Supplementary Data 1.

**Reporting summary**. Further information on research design is available in the Nature Research Reporting Summary linked to this article.

## Data availability

The datasets generated during and/or analyzed during the current study are available from the corresponding authors on request and are contained in the source data file. Source data are provided with this paper.

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

## Acknowledgements
We thank K. Regensburger, R. Burger, M. Meyer, J. Fischer, B. Ott, S. Engelhart, N. Schrott, N. Preising, and D. Krnavek for technical assistance. The ACE2 vector and the SARS-CoV-2 S-HA plasmid were provided by Shinji Makino and Stefan Pöhlmann. We thank K.-K. Conzelmann for providing VSVΔG and the Core Functional Peptidomics of Ulm University for peptide synthesis. This study was supported by DFG grants to F.K. (CRC 1279, SPP 1923), J.Mün. (CRC 1279), D.K. (KM 5/1-1), K.M.J.S. (CRC 1279, SPP 1923, SP1600/4-1, SP1600/6-1), D. Sa. (SPP 1923, Heisenberg-Program SA2676/3-1), C.G. (GO2153/3-1). EU's Horizon 2020 research and innovation program to J.M. (Fight-nCoV, 101003555), as well as the BMBF to F.K., D.Sa. and K.M.J.S. (Restrict SARS-CoV-2, protACT, and IMMUNOMOD), COVID-19 research grants from the Ministry of Science, Research and the Arts of Baden-Württemberg (MWK) to D.Sa., F.K. and J.Mün. and the Canon Foundation in Europe to D.Sa. and K.Sa., C.P.B., C.C., F.Z., L.K., T.W., L.W., J.K., D. Sch., F.D., and R.G. are part of and R.G. is funded by a scholarship from the International Graduate School in Molecular Medicine Ulm (IGradU).

## Author contributions
C.P.B. and R.N. performed most experiments. M.V. performed interaction assays. J.K., S.H., and A.K. generated and provided gut organoids. C.M.S. generated most expression constructs. D.K. performed MaMTH assays. J. Mül., C.C., and J. Mün prepared and provided SARS-CoV-2 stocks. F.Z. assisted in experiments with infectious SARS-CoV-2. L.W., T.W., K.S., and R.G. provided reagents and protocols. D. Sc. performed FACS for the Vpr-BlaM assay; E.B. and J.W. performed the HEK293T GFP split fusion assay. L.K. helped with the microscopy analysis of organoids and performed additional Western Blotting. F.D. and S.J. cultured and provided cardiomyocytes. A.C., M.S., and T.B. isolated and provided neurons. A.A.R.A. and S.W. analyzed peptides. D. Sa., C.G., K.S., S.S., and J. Mün provided comments and resources. K.M.J.S and F.K. conceived the study, planned experiments, and wrote the paper. All authors reviewed and approved the paper.

## Funding

## Competing interests
The authors declare no competing interests.
