## [Peer Review File · Nature Communications]

Reviewers' Comments:

Reviewer #1:

Remarks to the Author:

Bozzo et al. report a series of experiments that highlight a surprising role for cellular IFITM2 as a dependency factor for SARS-CoV-2 in lung cells. The authors perform a comprehensive study that includes elegant work with pseudoviruses and genuine SARS-CoV-2 in multiple cell lines and primary cells to identify that endogenous IFITMs do not inhibit, but rather promote, infection. This article is well written and the figures are nicely designed and well organized. Overall, the submission is important because a long list of recent publications have implicated the IFITM proteins in SARS-CoV-2 infection and COVID19 disease (and some of the most important are cited here by the authors), yet a mechanistic understanding of their involvement is lacking. Furthermore, the possibility that IFITM may be co-opted by SARS-CoV-2 (and other coronaviruses) is an intriguing one and this manuscripts offers a basis for this. I have several points that should be addressed in a revised version of the manuscript.

Major issues:

Figure 2: Since IFN-gamma did not increase IFITM2 levels in SAEC, while IFN-alpha and IFN-beta did, can the authors please test whether IFN-gamma inhibits SARS-CoV-2 infection of SAEC and perform Figure 2C in the presence of IFN-gamma?

Figure 2B: the blots do not indicate whether the specific IFITM1, IFITM2, or IFITM3 siRNAs are specific for one or more target. In other words, does siRNA for IFITM1 reduce IFITM1 as well as IFITM2 and IFITM3? The labeling needs to be modified so that is clear which siRNA corresponds to which blots and the authors should show data for the specificity of each siRNA by blotting for all three IFITMs following transfection of any individual siRNA. A clear demonstration of the specificity of the antibodies would assist in the readers' ability to interpret the results obtained using anti-IFITM antibodies to inhibit infection. Specificity should be established by citing and discussing previous reports which have done so and by the authors own efforts in house.

Line 152: The authors should perform the BLam assay on Calu3 or SAEC that have been treated with IFITM1, IFITM2, or IFITM3 siRNA?

Line 174: I do not agree that the switch from S/ACE2 interaction to S/RAB5A interaction signifies virus fusion in endosomes. It could simply signify the delivery of the virus-containing vesicle to an early endosomal compartment. RNAi of IFITM2 leads to a greater S/RAB5A interaction at 2h on ice, and so how do the authors interpret that? Is it evidence of increased fusion? Or evidence of a non-productive internalization pathway? If the authors believe fusion cannot occur at 4C, then what does S/RAB5A signify at 4C? Why do the authors choose RAB5A (an early endosome marker) rather than a late endosomal marker?

It would be really helpful to understand where IFITM2 binds Spike in terms of intracellular compartments. Can the authors costain with endosomal markers to highlight where the PLA signal is coming from? Does the interaction between Spike and IFITM2 require IFITM2 at the cell surface and its subsequent internalization? What would the PLA results look like if the authors used a mutant of IFITM2 (DeltaY19 or a version lacking the N-terminus) which is unable to be endocytosed?

Line 269: This statement is misleading. IFITM proteins are almost certainly involved in physiological functions other than antiviral activity, as highlighted in reference 14. One key example was detailed in the recent publication: Lee et al. IFITM3 functions as a PIP3 scaffold to amplify PI3K signalling in B cells, *Nature*, 2020. IFITM2 may perform similar functions in the PI3K pathway due to its sequence similarity to IFITM3. The authors should highlight the potential trade-offs of using anti-IFITM peptides in clinical settings.

Minor issues:

Some of the authors are not mentioned in the Contributions section. Furthermore, many authors are listed because of providing reagents and comments, which should not constitute authorship. Normally, these individuals would be listed in the Acknowledgements section.

The data in Figure 3D needs to be better described in the Results text so that readers can interpret the data in the figure. For example, what is "CTRL?" What does an increase in Gaussia activity indicate?

Figure 6B: the order of the conditions should be adjusted such that it matches 2C (anti-ACE2, anti-IFITM1-3, and mIFITM2 peptide).

Line 62: When referring to recent reports highlighting the importance of membrane rigidity and curvature to prevent fusion pore formation, please cite the recent Guo et al. *bioRxiv*, 2020 and Rahman et al. *eLife*, 2020.

Line 160: Instead of "3" spell out "IFITM3."

Line 165: The authors refer to Figure 2D but not such figure panel exists.

Line 224: The reference to results suggesting that peptide treatment mitigates the effects of virus on cardiomyocyte beating should be removed because that data is not shown.

Line 242: References are missing (Ref).

Reference 9 is duplicated in the bibliography. Reference 32 should be deleted and replaced with reference 9. Also, reference 33 is duplicated (reference 34 should be deleted).

Line 540: Did the versions of IFITM1, IFITM2, and IFITM3 expressed in pCG contain an epitope tag? It is hard to rationalize why ectopic expression and endogenous expression lead to opposite impacts on SARS-CoV-2 infection, especially since the authors claim it is not due to differential protein levels being expressed under ectopic and endogenous conditions.

Reviewer #2:

Remarks to the Author:

In this study, the authors investigate the role of the IFITM proteins in SARS-CoV-2 infection. Consistent with previous reports, they observe that ectopic overexpression of IFITM proteins inhibits the infectivity of pseudovirions bearing the SARS-CoV-2 spike (S) protein. In contrast, the endogenous expression of IFITM proteins appears to play a proviral role in the context of bona fide SARS-CoV-2 replication, as siRNA-mediated knock-down of these proteins (particularly IFITM1 or IFITM2, depending on the cell type) in human lung cells reduces virus replication. Consistent with a positive role for IFITM proteins in SARS-CoV-2 replication, peptides and antibodies targeting these proteins inhibit viral infection.

Previous studies with SARS-CoV-2 (e.g., Shi et al. *EMBO* 2020) and with a seasonal coronavirus have demonstrated that IFITM proteins can play proviral or antiviral roles depending on the specific circumstances; in this sense, the findings reported here are not entirely unprecedented. However, in my view, this study provides valuable new insights into the role of this important family of cellular proteins in coronavirus replication. The data are, for the most part, clear and convincing and the paper is well written for a broad audience. Specific issues to be addressed are listed below.

1. IFITM proteins have been reported to antagonize virus infectivity both when expressed in the virus-producer cell and when expressed in the target cell. The pseudotyped particle infectivity assays used in this study measure only the target-cell inhibition, whereas the assays performed with authentic SARS-CoV-2 virus are, in most cases, multi-round assays, or measure the infectivity of virus released into the supernatant. Thus the comparison of ectopic overexpression and endogenous expression using single-round or multi-round infection systems is a bit of an “apples-and-oranges” comparison. The authors need to discuss in more detail the producer vs target cell effects and address the implications for interpretation of their data. It would be useful to show data on the effect of knocking down the endogenous protein on pseudotype particle infectivity.
2. Fig. 1b. The authors use PSGL-1 as a control but don't mention this in the main text of the MS. The major inhibitory effect of PSGL-1 appears to be a result of its incorporation into virus particles and subsequent inhibition of particle binding to target cells. Thus its use in the target cell is somewhat unclear.
3. The authors show data (Fig. 3) suggesting that S protein interacts with IFITM proteins. They should mention that Shan-Lu Liu's group has reported a direct interaction between HIV-1 Env and IFITM proteins (Yu et al., Cell Rep 2015).
4. Fig. 4. I don't find these data as convincing as the other data in the paper. My recommendation would be to remove these results.
5. The paragraph beginning on line 154 mis-cites some of the figs (Fig. 2 instead of Fig. 3).
6. Line 242 and 267, references are missing.

Reviewer #3:

Remarks to the Author:

The manuscript by Bozzo CP et al. examined the role of IFITM proteins, a group of well-characterized ISGs in SARS-CoV-2 infection. Similar to other publications, the authors found that IFITM overexpression inhibited pseudotyped and authentic SARS-CoV-2 replication. The authors then made the surprising and interesting discovery that endogenous IFITMs supported SARS-CoV-2 infection, via binding to the spike protein and blockade of spike-ACE2 interaction. The manuscript provided a potentially important finding regarding SARS-CoV-2 hijacking host innate immunity for its own benefit. However, many pieces of the data were over-interpreted. Unfortunately, a number of key experiments also suffer from the lack of rigor (necessary controls and biological replicates).

My specific comments are as follows:

Major points:

- 1) The title of the paper does not fully reflect the results of the paper (endogenous vs inducible) and is misleading to readers (suggestive of IFITMs as potential therapeutic targets, which was not supported by the data).
- 2) The most important finding of the paper is presented in Fig. 1d, in which the siRNA knockdown of IFITMs led to reduced authentic SARS-CoV-2 infection. However, this contrasts the pseudotyped virus data in Fig. 1c. If the mechanism is as the authors elaborated in Fig. 3 and 4, i.e., via IFITM binding to spike and blocking ACE2 interaction, why is this specifically happening to the wild-type virus? This is also not reflected in the graphical summary in Extended data Fig. 10. It is possible that other SARS-CoV-2 proteins (i.e., M protein), not present in pseudotyped viruses, account for this discrepancy.
- 3) I do not find the mechanism data (Fig. 3 and 4) to be compelling. In the proximity ligation assay in Fig. 3b, what was the MOI used here? How could the authors detect spike proteins on incoming virus particles with host factors at 2 hr post infection? Also, fast forwarding to that in Fig. 4b, why was a different condition (2hr on ice and 15 min at 37C) used here to measure spike-ACE2 interaction?
- 4) Fig. 3e, for immunoprecipitation, the spike protein was expressed intracellularly, which is not in the context of SARS-CoV-2 entry. Other viral proteins such as M should be included as a proper control.
- 5) The flow cytometry data of surface IFITM staining in Extended data Fig. 7 is also not convincing. IFNs should be used as a positive control here. What is physiological relevance of Calu-3 cells that are

10-20% positive for IFITM1/2/3 (Fig. S7c)? If this is really the case, what is the viral infectivity in Fig. 1d and how could siRNA knockdown lead to a 10-fold reduction?

6) Statistics was missing from multiple figure panels, including but not limited to Fig. 2c, 3a, 4d, etc.

7) Several data points in Fig. 2c had huge error bars; others in Fig. 5b and 6d had only one data point. How many replicates were performed?

Minor points:

- 1) Line 56, a key reference: Shi G et al., EMBO J, 2020, which described opposing activities of IFITM proteins in SARS-CoV-2 infection, should be highlighted for detailed comparison, rather than getting buried in a series of papers here.
- 2) Line 65, multiple key references are missing. Buchrieser J et al., EMBO J, 2020; Zang R et al., PNAS, 2020; Winstone H et al., bioRxiv, 2020, all of which described anti-SARS-CoV-2 activities of IFITMs, should be cited here.
- 3) Does IFITM expression or knockdown affect surface ACE2 levels? From Extended data Fig. 1b, it seems that bulk ACE2 levels were reduced with higher IFITM expression.
- 4) Line 152, a QPCR based assay looking at viral RNA at 6 hr post infection cannot conclude that viral "entry" is impacted here.
- 5) The signals should be enhanced for un-zoomed in images in Fig. 3 and 4.
- 6) Line 242 and 267, references are missing.
- 7) What is the efficacy of ectodomain of IFITM on an Fc stem in Fig. 5?
- 8) Fig. 6c is of poor quality and should be improved.

Reply to the reviewer`s comments (in *italic* letters)

Reviewer #1: Bozzo et al. report a series of experiments that highlight a surprising role for cellular IFITM2 as a dependency factor for SARS-CoV-2 in lung cells. The authors perform a comprehensive study that includes elegant work with pseudoviruses and genuine SARS-CoV-2 in multiple cell lines and primary cells to identify that endogenous IFITMs do not inhibit, but rather promote, infection. This article is well written and the figures are nicely designed and well organized. Overall, the submission is important because a long list of recent publications have implicated the IFITM proteins in SARS-CoV-2 infection and COVID19 disease (and some of the most important are cited here by the authors), yet a mechanistic understanding of their involvement is lacking. Furthermore, the possibility that IFITM may be co-opted by SARS-CoV-2 (and other coronaviruses) is an intriguing one and this manuscripts offers a basis for this. I have several points that should be addressed in a revised version of the manuscript.

We are pleased that reviewer 1 feels that our manuscript is “well written” and organized as well as “comprehensive” and “important”. As outlined below, we addressed all issues raised.

1. Figure 2: Since IFN-gamma did not increase IFITM2 levels in SAEC, while IFN-alpha and IFN-beta did, can the authors please test whether IFN-gamma inhibits SARS-CoV-2 infection of SAEC and perform Figure 2C in the presence of IFN-gamma?

We performed this experiment. The results confirmed that KD of IFITM1 reduces virus production and further showed that KD of IFITMs had no additional inhibitory effect in the presence of IFN-gamma (new Fig. 2C; new Supplementary Fig. 6b, lines 133-135). This result agrees with our finding that IFN-gamma has only modest effects on IFITM expression levels in SAEC cells (Fig. 2a).

2. Figure 2B: the blots do not indicate whether the specific IFITM1, IFITM2, or IFITM3 siRNAs are specific for one or more target. In other words, does siRNA for IFITM1 reduce IFITM1 as well as IFITM2 and IFITM3? The labeling needs to be modified so that is clear which siRNA corresponds to which blots and the authors should show data for the specificity of each siRNA by blotting for all three IFITMs following transfection of any individual siRNA. A clear demonstration of the specificity of the antibodies would assist in the readers' ability to interpret the results obtained using anti-IFITM antibodies to inhibit infection. Specificity should be established by citing and discussing previous reports which have done so and by the authors own efforts in house.

Reviewer 1 raises important points. To address them, we examined the specificity of the antibodies and siRNAs. We show that Abs against IFITM1 and IFITM3 are specific, despite the high sequence identity of the three IFITMs (new Supplementary Fig. 1c). The antibody raised against IFITM2 also recognizes IFITM3, albeit with reduced efficiency (new Supplementary Fig. 1c). This is expected since both are highly similar. To assess siRNA specificity, while avoiding antibody bias, we generated and tested qPCR primer/probe sets specific for the three different IFITM mRNAs (new Supplementary Fig. 3b). Our results indicate that IFITM siRNAs are specific for their respective targets (new Supplementary Fig. 3c). However, as mentioned in the revised manuscript (lines 102-106) low cross-silencing was observed for siRNAs against IFITM2 and IFITM3. Notably, the importance of IFITM2 for SARS-CoV-2 infection is also supported by IFITM-derived peptides and Spike-interaction studies. Most importantly, cross-reactivity of the IFITM2 Ab with IFITM3 does not compromise the main conclusions of our study.

3. Line 152: The authors should perform the BLam assay on Calu3 or SAEC that have been treated with IFITM1, IFITM2, or IFITM3 siRNA?

As suggested, we performed the BLam fusion assay of Calu-3 cells treated with control or IFITM-targeting siRNA and exposed to HIV_Blam-Vpr_SARS-CoV-2-Spike pseudotypes for 2 hours (see below, Figure 1). There was a modest non-significant reduction upon IFITM2 siRNA treatment. Altogether, however, the levels of fusion were relatively low and highly variable in independent repeats. Since Spike-pseudotyped lentiviral particles are

relatively artificial, we decided not to pursue it further but to focus on genuine SARS-CoV-2 in most experiments.

Figure 1: Fusion of HIV(Vpr-Blam) Δ env*-SARS-CoV-2-S with Calu-3 depleted of indicated IFITMs. Quantification of the fusion efficiency by flow cytometry as percentage of (cleaved CCF2) positive cells. Bars represent means (\pm SEM) of two experiments each done in technical duplicates.

4. Line 174: I do not agree that the switch from S/ACE2 interaction to S/RAB5A interaction signifies virus fusion in endosomes. It could simply signify the delivery of the virus-containing vesicle to an early endosomal compartment. RNAi of IFITM2 leads to a greater S/RAB5A interaction at 2h on ice, and so how do the authors interpret that? Is it evidence of increased fusion? Or evidence of a non-productive internalization pathway? If the authors believe fusion cannot occur at 4C, then what does S/RAB5A signify at 4C? Why do the authors choose RAB5A (an early endosome marker) rather than a late endosomal marker?

Our results suggest that some low-level transport of virus-containing vesicles and subsequent fusion may already occur at 4 °C. Viral fusion would lead to a reduction of S/ACE2 signals. Our interpretation is that increased S/Rab5 proximity at 4°C after IFITM2 depletion indicates enhanced accumulation of viral particles in early endosomes due to a further reduction of the background levels of fusion. The number of S/RAB5A signals rapidly increases and ACE2/S signals strongly decreased after shifting the cells to 37 °C (Figs. 4c, 4d). These results agree with our hypothesis that IFITM2 interacts with the Spike protein to promote S/ACE2-mediated fusion in early endosomes and ultimately enhances SARA CoV-2 production. We used a marker for early endosomes because it has been reported that SARS-CoV-2 may fuse at the cell surface as well as in early endosomes (e.g. Shang et al., PNAS 2020) and it has been shown that IFITM2 is localized predominantly to early endosomes, while IFITM1 and IFITM3 localize to the plasma membrane and late endosomes, respectively (e.g. Narayana et al., JBC 2015). As outlined in the reply to the next point raised by reviewer 1, we obtained further evidence for S/IFITM2 interaction in early endosomes (new Fig. 4e-g, lines 198-202). However, since the indication for fusion in endosomes is indirect, we cautioned our conclusions throughout.

5. It would be really helpful to understand where IFITM2 binds Spike in terms of intracellular compartments. Can the authors costain with endosomal markers to highlight where the PLA signal is coming from? Does the interaction between Spike and IFITM2 require IFITM2 at the cell surface and its subsequent internalization? What would the PLA results look like if the authors used a mutant of IFITM2 (DeltaY19 or a version lacking the N-terminus) which is unable to be endocytosed?

We thank the reviewer for these helpful suggestions and were able to perform the proposed technically challenging experiments. Combining PLA with concurrent antibody staining, we clearly show that Spike interacts with IFITM2 in early endosomes (new Figs. 4e-g; new Supplementary Movies 1 and 2; see lines 198-202). We now also show that DeltaY19 mutation significantly reduces number of IFITM2/Spike PLA signals (new Supplementary Fig. 8b, lines 174-179). The background in this experimental setting was relatively high due to endogenous IFITM2 expression. Nonetheless, the results clearly support that proper localization and trafficking of IFITM plays a role in S/IFITM2 interaction.

6. Line 269: This statement is misleading. IFITM proteins are almost certainly involved in physiological functions other than antiviral activity, as highlighted in reference 14. One key example was detailed in the recent publication: Lee et al. IFITM3 functions as a PIP3 scaffold to amplify PI3K signalling in B cells, Nature, 2020. IFITM2 may perform similar functions in the PI3K pathway due to its sequence similarity to IFITM3. The authors should highlight the potential trade-offs of using anti-IFITM peptides in clinical settings.

We agree and modified the text and references accordingly (lines 305-309).

Minor issues:

Some of the authors are not mentioned in the Contributions section. Furthermore, many authors are listed because of providing reagents and comments, which should not constitute authorship. Normally, these individuals would be listed in the Acknowledgements section.

We updated the contribution section and now mention all authors. The reviewer is correct that the list of coauthors is comprehensive. We started research on SARS-CoV-2 just recently and the generation of most reagents required for this study involve the establishment of novel methods. Thus, we feel that the inclusion of people providing important resources and critical unpublished reagents and protocols warrants coauthorship in this case.

The data in Figure 3D needs to be better described in the Results text so that readers can interpret the data in the figure. For example, what is “CTRL?” What does an increase in Gaussia activity indicate?

We now describe these results in more details (lines 183-187). In brief, an increase in Gaussia activity indicates interaction of the bait (here the Spike) and Prey (here the IFITMs) at the cell membrane. As mentioned in the methods (lines 615/616) the Gal4 (transcription factor) Bait with SHC1 Prey served as positive control (CTRL).

Figure 6B: the order of the conditions should be adjusted such that it matches 2C (anti-ACE2, anti-IFITM1-3, and mIFITM2 peptide).

The figure has been modified as suggested.

Line 62: When referring to recent reports highlighting the importance of membrane rigidity and curvature to prevent fusion pore formation, please cite the recent Guo et al. bioRxiv, 2020 and Rahman et al. eLife, 2020.

These studies are now cited.

Line 160: Instead of “3” spell out “IFITM3.”

Done.

Line 165: The authors refer to Figure 2D but not such figure panel exists.

Corrected.

Line 224: The reference to results suggesting that peptide treatment mitigates the effects of virus on cardiomyocyte beating should be removed because that data is not shown.

Done.

Line 242: References are missing (Ref).

Added.

Reference 9 is duplicated in the bibliography. Reference 32 should be deleted and replaced with reference 9. Also, reference 33 is duplicated (reference 34 should be deleted).

We thank reviewer 1 for making us aware of this and corrected the references.

Line 540: Did the versions of IFITM1, IFITM2, and IFITM3 expressed in pCG contain an epitope tag? It is hard to rationalize why ectopic expression and endogenous expression lead to opposite impacts on SARS-CoV-2 infection, especially since the authors claim it is not due to differential protein levels being expressed under ectopic and endogenous conditions.

The IFITMs expressed by pCG did not contain a tag. It is known that ectopic expression might lead to altered localization. Our FACS analyses suggest that e.g. a higher proportion of IFITM2 may be accessible at the surface of IFN β -treated Calu-3 cells compared to transfected HEK293T cells (revised or new Figs. supplementary Figs. 10 and 11). In addition, the IFITM1-3 mAb recognized IFITMs expressed by Calu-3 cells more efficiently than those expressed by HEK293T cells. Notably, the exact topology of IFITMs is unclear and there might be differences between ectopic and endogenous IFITMs. We now briefly discuss this in the revised manuscript (lines 286-290).

Reviewer #2 (Remarks to the Author):

In this study, the authors investigate the role of the IFITM proteins in SARS-CoV-2 infection. Consistent with previous reports, they observe that ectopic overexpression of IFITM proteins inhibits the infectivity of pseudovirions bearing the SARS-CoV-2 spike (S) protein. In contrast, the endogenous expression of IFITM proteins appears to play a proviral role in the context of bona fide SARS-CoV-2 replication, as siRNA-mediated knock-down of these proteins (particularly IFITM1 or IFITM2, depending on the cell type) in human lung cells reduces virus replication. Consistent with a positive role for IFITM proteins in SARS-CoV-2 replication, peptides and antibodies targeting these proteins inhibit viral infection.

Previous studies with SARS-CoV-2 (e.g., Shi et al. EMBO 2020) and with a seasonal coronavirus have demonstrated that IFITM proteins can play proviral or antiviral roles depending on the specific circumstances; in this sense, the findings reported here are not entirely unprecedented. However, in my view, this study provides valuable new insights into the role of this important family of cellular proteins in coronavirus replication. The data are, for the most part, clear and convincing and the paper is well written for a broad audience. Specific issues to be addressed are listed below.

We are pleased that reviewer 2 feels that our study “provides valuable new insights” and is largely “clear and convincing”. As noted by this reviewer it is interesting that overexpression of IFITM3 has been shown to promote infection by hCoV-OC43, which utilizes a different receptor. Notably, the Shi et al. study reported enhancing effects only for specific mutant forms of IFITM3 and concluded that “we have identified IFITM1, IFITM2, and IFITM3 as restrictors of SARS-CoV-2 infection of cells”. Notably, none of these studies examine the effect of endogenously expressed IFITMs on genuine virus infection. Thus, our finding that IFITMs are hijacked for efficient SARS-CoV-2 infection under physiological conditions is new.

1. IFITM proteins have been reported to antagonize virus infectivity both when expressed in the virus-producer cell and when expressed in the target cell. The pseudotyped particle infectivity assays used in this study measure only the target-cell inhibition, whereas the assays performed with authentic SARS-CoV-2 virus are, in most cases, multi-round assays, or measure the infectivity of virus released into the supernatant. Thus the comparison of ectopic overexpression and endogenous expression using single-round or multi-round infection systems is a bit of an “apples-and-oranges” comparison. The authors need to discuss in more detail the producer vs target cell effects and address the implications for interpretation of their data. It would be useful to show data on the effect of knocking down the endogenous protein on pseudotype particle infectivity.

We agree that it is important to consider different assay conditions and had analyzed the impact of IFITMs expression in the producer cells on the infectiousness of Spike VSVpp. To address this reviewer’s concern, we included these data in the revised manuscript (new Supplementary Fig. 2; lines 92-98). To verify the effect of IFITMs on target-cell infection of genuine SARS-CoV-2, we measured the levels of intracellular RNA just 6 hours after virus exposure (Fig. 2d). This basically excludes multiple rounds of replication. However, we found that silencing of IFITM expression usually affected the infectious titers more strongly than the viral RNA yields. We

now discuss in more detail that IFITMs might be relevant in both the producer as well as the target cells (lines 267-271).

2. Fig. 1b. The authors use PSGL-1 as a control but don't mention this in the main text of the MS. The major inhibitory effect of PSGL-1 appears to be a result of its incorporation into virus particles and subsequent inhibition of particle binding to target cells. Thus, its use in the target cell is somewhat unclear.

Efficient inhibition of SARS-CoV-2 infection by expression of PSGL-1 in the producer cells was indeed unexpected and is certainly of interest for follow-up studies. We did not observe an inhibitory effect of PSGL-1 on VSV-G pseudotypes. Steric hindrance has been reported as one potential antiviral mechanism, which may also be effective if PSGL-1 is expressed in the target cells. To streamline the manuscript and remove distraction from the main focus of our study, we omitted the PSGL-1 results from the revised manuscript.

3. The authors show data (Fig. 3) suggesting that S protein interacts with IFITM proteins. They should mention that Shan-Lu Liu's group has reported a direct interaction between HIV-1 Env and IFITM proteins (Yu et al., Cell Rep 2015).

We now mention this finding and cite this manuscript (lines 292-293).

4. Fig. 4. I don't find these data as convincing as the other data in the paper. My recommendation would be to remove these results.

We carefully considered this but decided to keep these results. These experiments are technically demanding and provide first insights into the possible mechanisms. Notably, reviewer 1 asked us to expand these analyses and our new results (new Fig. 4e-g) clearly support an interaction of the Spike protein with IFITM2 in early endosomes.

5. The paragraph beginning on line 154 mis-cites some of the figs (Fig. 2 instead of Fig. 3).

Corrected.

6. Line 242 and 267, references are missing.

Added.

Reviewer #3 (Remarks to the Author):

The manuscript by Bozzo CP et al. examined the role of IFITM proteins, a group of well-characterized ISGs in SARS-CoV-2 infection. Similar to other publications, the authors found that IFITM overexpression inhibited pseudotyped and authentic SARS-CoV-2 replication. The authors then made the surprising and interesting discovery that endogenous IFITMs supported SARS-CoV-2 infection, via binding to the spike protein and blockade of spike-ACE2 interaction. The manuscript provided a potentially important finding regarding SARS-CoV-2 hijacking host innate immunity for its own benefit. However, many pieces of the data were over-interpreted. Unfortunately, a number of key experiments also suffer from the lack of rigor (necessary controls and biological replicates).

We thank this reviewer for appreciating that our discovery is "surprising and interesting". However, we feel that all experiments were well controlled and interpreted with caution. As specified below, some previous experiments included averages of four independent biological replicates and not single data points or technical replicates. In addition, we performed additional repeats to further substantiate our data.

Major points:

1) The title of the paper does not fully reflect the results of the paper (endogenous vs inducible) and is misleading to readers (suggestive of IFITMs as potential therapeutic targets, which was not supported by the data).

We disagree. Endogenous and not artificially overexpressed IFITMs are physiologically relevant. Thus, we feel that it is appropriate to keep the present title. Our results demonstrate that antibodies against IFITMs and IFITM2-derived peptides reduce SARS-CoV-2 production by human lung cells, cardiomyocytes and gut organoids by up to two orders of magnitude. The effects on infectious virus yields as determined in plaques assays were even stronger. Thus, the conclusion that IFITMs represent a target for inhibition of SARS-CoV-2 is warranted.

2) The most important finding of the paper is presented in Fig. 1d, in which the siRNA knockdown of IFITMs led to reduced authentic SARS-CoV-2 infection. However, this contrasts the pseudotyped virus data in Fig. 1c. If the mechanism is as the authors elaborated in Fig. 3 and 4, i.e., via IFITM binding to spike and blocking ACE2 interaction, why is this specifically happening to the wild-type virus? This is also not reflected in the graphical summary in Extended data Fig. 10. It is possible that other SARS-CoV-2 proteins (i.e., M protein), not present in pseudotyped viruses, account for this discrepancy.

Notably, the finding that IFITMs promote genuine SARS-CoV-2 infection in Fig. 1d is also supported by several additional lines of evidence, e.g. by the inhibitory effect of IFITM Abs and peptides. While our assays indicate that Spike interacts with IFITMs this does by no means imply that it competes with ACE2-Spike interaction. Preliminary results of in silico interaction analyses suggest that different regions of the Spike protein interact with IFITMs and ACE2. While further work is required to fully elucidate the molecular details, the most plausible scenario is that IFITMs are hijacked by Spike to promote virus attachment, delivery into early endosomes and subsequent fusion between the viral and cellular membranes. The reasons why this enhancing effect is specific for genuine SARS-CoV-2 and not for Spike pseudotypes may include different membrane composition and curvature as well as the presence of other viral proteins as suggested by the reviewer (see lines 286-297).

3) I do not find the mechanism data (Fig. 3 and 4) to be compelling. In the proximity ligation assay in Fig. 3b, what was the MOI used here? How could the authors detect spike proteins on incoming virus particles with host factors at 2 hr post infection? Also, fast forwarding to that in Fig. 4b, why was a different condition (2hr on ice and 15 min at 37C) used here to measure spike-ACE2 interaction?

The MOI in these experiments was 1.5 to ensure sufficient amount of virus. The PLA assay is sensitive enough to detect Spike proteins on incoming, bound or fusing SARS-CoV-2 particles that are in close proximity to IFITMs. The results agree well with those from Co-IP and other assays supporting that the Spike protein interacts with IFITM2 on Calu-3 cells. Incubation for 2h on ice is classically used to allow initial attachment but suppress subsequent transport and/or fusion. This is triggered by raising the temperature to 37C. Thus, the experiments shown in Fig. 4 were designed to obtain insights into the dynamics of endosomal delivery and the fusion process. Notably, our new data (new Fig. 4e-g) support the initial conclusions.

4) Fig. 3e, for immunoprecipitation, the spike protein was expressed intracellularly, which is not in the context of SARS-CoV-2 entry. Other viral proteins such as M should be included as a proper control.

Viral glycoproteins may already interact with their receptors during transport to the cell surface. In addition, Spike protein present at the cell surface may bind to IFITMs on neighboring cells. Finally, during a co-immunoprecipitation the cells are lysed allowing proteins to interact. Viral matrix (M) proteins are frequently "sticky" and showing non-specific interactions. Thus, we confirmed the specificity of the interactions between Spike and IFITMs using the Nsp7 protein as an additional control (Figure 2).

Figure 2: HEK 293T cells were transfected with Spike or Nsp7, together with IFITM1, IFITM2 or IFITM3. 24h later cells were lysed and lysates were incubated with the magnetic beads bound to V5 antibody to precipitate Spike or Nsp7.

5) The flow cytometry data of surface IFITM staining in Extended data Fig. 7 is also not convincing. IFNs should be used as a positive control here. What is physiological relevance of Calu-3 cells that are 10-20% positive for IFITM1/2/3 (Fig. S7c)? If this is really the case, what is the viral infectivity in Fig. 1d and how could siRNA knockdown lead to a 10-fold reduction?

We performed the FACS analysis in Calu-3 cells and clearly demonstrate that treatment with IFN increases the expression of surface-accessible IFITMs as expected from the western blot analyses (new Supplementary Fig. 10d-f). SARS-CoV-2 may preferentially infect cells expressing IFITMs as well as ACE2 and TMPRSS2. It is conceivable that infection efficiency is drastically reduced if one important entry factor is not available.

6) Statistics was missing from multiple figure panels, including but not limited to Fig. 2c, 3a, 4d, etc.

We added statistics and additional repeats wherever appropriate. Exact p-values for all figures are now provided in a supplemental table. Our key results, i.e. the dependency of SARS-CoV-2 on IFITM2 for efficient infection of Calu-3 cells and the inhibitory effects of IFITM Abs or peptides were highly significant and confirmed in a variety of different experimental settings.

7) Several data points in Fig. 2c had huge error bars; others in Fig. 5b and 6d had only one data point. How many replicates were performed?

The effects shown in Fig. 2c were highly reproducible. In some experiments, virus production was reduced to background levels in the presence of IFN- β upon KD of IFITM proteins. Fig. 5b initially showed the results of two independent experiments each performed in duplicate, it was repeated and proper statistics are now provided. Each datapoint in Fig. 6d (now 6b), represents the average of four measurements (SDs are indicated but were apparently missed by this reviewer). Altogether, viral RNA yields were measured at five time points after infection with 5 different MOIs. Altogether, this figure is based on a total of ~120 viral RNA measurements.

Minor points:

1) Line 56, a key reference: Shi G et al., EMBO J, 2020, which described opposing activities of IFITM proteins in SARS-CoV-2 infection, should be highlighted for detailed comparison, rather than getting buried in a series of papers here.

This title of this study is somewhat misleading since significant enhancing effects were only observed for specific mutant forms of IFITM3. The request to highlight a specific paper for detailed comparison seems unusual. While we cite it, we did not discuss it in detail to keep our manuscript clear and concise. Notably, the authors of this study did not cite or discuss our preprint although they made positive comments about it on BioRxiv well before they published their study.

2) Line 65, multiple key references are missing. Buchrieser J et al., EMBO J, 2020; Zang R et al., PNAS, 2020; Winstone H et al., bioRxiv, 2020, all of which described anti-SARS-CoV-2 activities of IFITMs, should be cited here.

As suggested by the reviewer, in the revised version we cite additional studies that seem relevant. Notably, to our knowledge none of them analyzed the impact of endogenous IFITM expression on authentic SARS-CoV-2.

3) Does IFITM expression or knockdown affect surface ACE2 levels? From Extended data Fig. 1b, it seems that bulk ACE2 levels were reduced with higher IFITM expression.

We occasionally observed modest effects on ACE2 expression in overexpression assay (see for example Supplementary Fig. 1b). Further examination showed that IFITM KD had not significant effect on ACE2 expression levels (new Supplementary Fig.3d).

4) Line 152, a QPCR based assay looking at viral RNA at 6 hr post infection cannot conclude that viral “entry” is impacted here.

We rephrased the conclusion to “an early step of infection” (line 165-167).

5) The signals should be enhanced for un-zoomed in images in Fig. 3 and 4.

Done

6) Line 242 and 267, references are missing.

Added.

7) What is the efficacy of ectodomain of IFITM on an Fc stem in Fig. 5?

We changed the labeling for clarity.

8) Fig. 6c is of poor quality and should be improved.

The quality has been updated to 300 dpi.

Reviewers' Comments:

Reviewer #1:

Remarks to the Author:

In this revised manuscript, Bozzo et al. have responded to concerns and introduced new data for consideration by reviewers. There remain a few points of contention that need to be addressed.

Major issues:

In Figure 1, supernatants from Calu3 cells treated with siRNA and exposed to SARS-CoV-2 are added to Vero cells. This experiment is much appreciated, as it's one of the only places in the manuscript where infectious virus is assessed. However, the results in Figure 1E should be quantified from multiple experiments. The cytopathic effect on Vero cells does not match up well with the RNA counts in Calu3, indicating that there is a disconnect between measuring RNA and infectious virus yields. A focus on SARS-CoV-2 RNA levels in most figures may be overestimating the impact of endogenous IFITMs (especially IFITM2) on infection.

Given the importance of the findings using an anti-IFITM2 antibody as a potential therapeutic, the authors should use at least one other anti-IFITM2. This is especially important because the authors show that their anti-IFITM2 antibody recognizes both IFITM2 and IFITM3, almost equally. I suggest that the authors use the following anti-IFITM2, because it was shown to be exquisitely specific: anti-IFITM2 (66137-1-Ig; Proteintech; specificity demonstrated in Shi et al. PNAS 2018).

The new results added to the PLA assay using the DeltaY19 IFITM2 mutant do not conform to a model whereby surface associated IFITM2 acts as a dependency factor for SARS-CoV-2. The DeltaY19 mutant is predominantly expressed at the cell surface and/or in vesicular structures near the cell surface (i.e. early endosomes), and so why this mutant does not result in an enhancement of IFITM2-Spike binding (relative to WT IFITM2) is unclear. These data actually suggest that the IFITM2-Spike interaction occurs during the act of virus endocytosis, and the text should be modified to better reflect that. But if that is the case, it's hard to understand how the cell surface staining of IFITM proteins in cells contributes to our interpretation of the results. The reviewers raised concerns about the use of this PLA data to guide mechanistic explanations for how IFITM proteins impact SARS-CoV-2 infection, and this is especially the case because the PLA was performed in HeLa-ACE2 cells. There is no indication that knockdown of IFITMs in HeLa-ACE2 cells promotes SARS-CoV-2 infection. Since IFITM proteins have been shown to traffic to incoming Influenza A Virus-containing vesicles (Spence et al.), it is possible that the authors are actually seeing an increase in IFITM/Spike colocalization because IFITM proteins are engaging the early endosomes which contain virus particles. The authors claim that their use of lung cells is what allowed them to discover IFITM proteins as host dependency factors for SARS-CoV-2, yet lung cells were not used in the PLA.

Another concern is that the authors provide no compelling argument for why ectopic IFITM expression does not recapitulate the effects of endogenous IFITM proteins on SARS-CoV-2 expression, even when the levels of ectopic protein are titered from very low amounts to larger amounts. It is possible that the amino acid sequence encoded by their overexpression plasmids does not match the sequence of endogenous IFITM in the cells/tissues they are studying. Is it possible that the cells/tissues they are working with express different isoforms of IFITMs which are not produced from the overexpression constructs, or do the cell lines express endogenous IFITM proteins encoding mutations? Another possibility is that ectopic IFITM expression interferes with whatever endogenous IFITM protein is expressed in a given cell line, by disrupting homooligomerization or heteromultimerization among IFITM family members, or by disrupting an interaction between endogenous IFITM and a non-IFITM cofactor. The suggestion by the authors that IFITM proteins could be targeted as antiviral therapy is premature and unrealistic, especially considering the tradeoff that this would present for resistance to other virus infections, like Influenza A Virus. Since IFITM proteins facilitate PI3K signaling, is it possible that IFITM-derived peptides would inhibit cell growth and promote apoptosis? This might be

desirable in the setting of cancer, but perhaps not in healthy individuals trying to stave off virus infections.

Minor issues:

Line 291: text should be changed to "The antiviral activity of IFITMs is very broad and does not usually not involve interactions with specific viral glycoproteins."

When citing reference 21, there are a host of other papers that could be cited. From 2014 to 2021, there are now several studies on the effects of IFITM proteins in virus-producing cells. I would recommend citing a few others.

In Figure 6, the right side of 6A does not appear to be described in the Figure legend. What does "mIFITM2 peptide" signify? This reagent is not adequately described in the text, figure legend, or methods section. It also appears in Supplemental Figure 14 but is not described. The human-derived peptides are clearly described in Figure 5 but I'm not seeing a similar indication of what mIFITM2 is.

Reviewer #2:

Remarks to the Author:

the authors have addressed my concerns.

Reviewer #3:

Remarks to the Author:

This is a revision of the manuscript by Bozzo CP et al. The authors have made extensive revision and added new data (and replicates) to address the potential concerns. In particular, new Fig. 4 e-g provides convincing proof of IFITM2-spike interactions and this likely takes place at the early endosomes. However, this reviewer still has significant concerns over the flow cytometry data in Supplementary Fig. 10. Comparing panels a and d, The newly carried out IFN- β treatment altered the PE staining pattern, even for the isotype controls. IFN- β also did not seem to enhance intracellular IFITM2 levels. Does siRNA transfection lead to a specific reduction of surface IFITM2 in Calu3 cells? If the authors really believe that SARS-CoV-2 preferentially infects IFITM-positive cells (as is stated in the rebuttal letter), then such data should be added to the paper. Otherwise, I suggest simply removing the flow cytometry data to avoid confusion.

Reply to the reviewers (in *italic* letters)

Reviewer #1: In this revised manuscript, Bozzo et al. have responded to concerns and introduced new data for consideration by reviewers. There remain a few points of contention that need to be addressed.

We appreciate that reviewer 1 acknowledges that we addressed all previous concerns by adding new data but were surprised that new issues are raised. The remaining “points of contention” seem to result largely from misinterpretation of our data.

In Figure 1, supernatants from Calu3 cells treated with siRNA and exposed to SARS-CoV-2 are added to Vero cells. This experiment is much appreciated, as it's one of the only places in the manuscript where infectious virus is assessed. However, the results in Figure 1E should be quantified from multiple experiments. The cytopathic effect on Vero cells does not match up well with the RNA counts in Calu3, indicating that there is a disconnect between measuring RNA and infectious virus yields. A focus on SARS-CoV-2 RNA levels in most figures may be overestimating the impact of endogenous IFITMs (especially IFITM2) on infection.

To address this issue, we determined the infectious virus titers using both plaque assays and TCID50 assays (new Figs. 1e, 1g, S4d, S4e). Results were highly reproducible and the infectious titers correlated very well with the viral RNA measurements (new Fig. 1h, 1i). In agreement with our previous studies (Nchioua et al., mBio 2020; Hayn et al., Cell Reports 2021), viral RNA measurements by qPCR under- instead of overestimate effects on infectious virus loads.

Given the importance of the findings using an anti-IFITM2 antibody as a potential therapeutic, the authors should use at least one other anti-IFITM2. This is especially important because the authors show that their anti-IFITM2 antibody recognizes both IFITM2 and IFITM3, almost equally. I suggest that the authors use the following anti-IFITM2, because it was shown to be exquisitely specific: anti-IFITM2 (66137-1-Ig; Proteintech; specificity demonstrated in Shi et al. PNAS 2018).

We did not find information which domain of IFITM2 is targeted by the proposed Ab as it was just used for western blot analyses of IFITM in denatured, non-native conformation in the Shi et al. study. In contrast to the inhibitory Abs used in our study, the Shi et al Ab does not recognize the surface exposed epitope of native IFITM2 (Fig. R1). Thus, it is not useful for inhibition studies. Notably, we already showed that treatment with two different antibodies (Figs. 5b, 6a, S14) and an IFITM2-derived peptide (Figs. 5c, 6a, 6d, S14) as well as IFITM2 kd (Figs. 1d, 1e, 2b, 2d, S4, S6) inhibit SARS-CoV-2.

Fig. R1: Flow cytometric analysis of IFITM expression in Calu-3 cells comparing the IFITM2 Cell Signaling antibody (orange) with the suggested IFITM2 Proteintech (blue) antibody. Gating strategy and flow cytometric detection of endogenous IFITMs in non-permeabilized (upper panel) or permeabilized (lower panel) Calu-3 cells using indicated α -IFITM2 antibodies. Examples of primary FACS data indicating the gating strategy. (right panels) Histograms showing mean fluorescence intensities of stained cells normalized for alive/single cells. (bottom panels) Bars represent the mean percentage of positively stained cells, $n=2$ (independent replicates, \pm SEM).

The new results added to the PLA assay using the DeltaY19 IFITM2 mutant do not conform to a model whereby surface associated IFITM2 acts as a dependency factor for SARS-CoV-2. The DeltaY19 mutant is predominantly expressed at the cell surface and/or in vesicular structures near the cell surface (i.e. early endosomes), and so why this mutant does not result in an enhancement of IFITM2-Spike binding (relative to WT IFITM2) is unclear. These data actually suggest that the IFITM2-Spike interaction occurs during the act of virus endocytosis, and the text should be modified to better reflect that. But if that is the case, it's hard to understand how the cell surface staining of IFITM proteins in cells contributes to our interpretation of the results. The reviewers raised concerns about the use of this PLA data to guide mechanistic explanations for how IFITM proteins impact SARS-CoV-2 infection, and this is especially the case because the PLA was performed in HeLa-ACE2 cells. There is no indication that knockdown of IFITMs in HeLa-ACE2 cells promotes SARS-CoV-2 infection. Since IFITM proteins have been shown to traffic to incoming Influenza A Virus-containing vesicles (Spence et al.), it is possible that the authors are actually seeing an increase in IFITM/Spike colocalization because IFITM proteins are engaging the early endosomes which contain virus particles. The authors claim that their use of lung cells is what allowed them to discover IFITM proteins as host dependency factors for SARS-CoV-2, yet lung cells were not used in the PLA.

Contrary to the statement above, most PLA assays were actually performed in human lung cells infected with genuine SARS-CoV-2 (Figs. 3a, 3b, 4a-g). Only the experiment previously requested by reviewer 1 involving transient expression of mutant IFITM (Fig. S8b) was performed in HeLa cells since lung cells could not be efficiently transfected. Even the previously requested, technically demanding simultaneous PLA and co-staining of endosomal markers (Fig. 4e-g) were performed in human lung cells.

All our results are consistent and indicate that SARS-CoV-2 targets IFITMs already at the cell surface but fusion largely occurs in early endosomes. This also explains why the mutant IFITM in HeLa cells show reduced functional activity as it is not endocytosed. Notably, this data and our interpretation of the underlying mechanism convinced the other reviewers. We introduced some textual changes to further increase the clarity of the manuscript.

Another concern is that the authors provide no compelling argument for why ectopic IFITM expression does not recapitulate the effects of endogenous IFITM proteins on SARS-CoV-2 expression, even when the levels of ectopic protein are titrated from very low amounts to larger amounts. It is possible that the amino acid sequence encoded by their overexpression plasmids does not match the sequence of endogenous IFITM in the cells/tissues they are studying. Is it possible that the cells/tissues they are working with express different isoforms of IFITMs which are not produced from the overexpression constructs, or do the cell lines express endogenous IFITM proteins encoding mutations? Another possibility is that ectopic IFITM expression interferes with whatever endogenous IFITM protein is expressed in a given cell line, by disrupting homooligomerization or heteromultimerization among IFITM family members, or by disrupting an interaction between endogenous IFITM and a non-IFITM cofactor.

As mentioned in the discussion section (lines 291-303) there are numerous possible reasons (structure, localization, trafficking, ...) why artificially overexpressed IFITMs may not recapitulate the function of endogenous IFITMs, some given by the reviewer in the comment. The plasmids used were sequenced and match the correct IFITM sequences. The enhancing effects involve specific interactions of the viral Spike protein with the N-terminal region of IFITM. In contrast, the inhibitory activity doesn't seem to involve specific interactions with viral envelope proteins. Altogether, we present various lines of evidence in several cell lines, primary cells and organoids that endogenously expressed IFITMs act as SARS-CoV-2 entry cofactors in physiologically relevant settings. Determining the reason why IFITMs have the opposite effect under artificial conditions is beyond the scope of the present already very comprehensive study and clearly not relevant.

The suggestion by the authors that IFITM proteins could be targeted as antiviral therapy is premature and unrealistic, especially considering the tradeoff that this would present for resistance to other virus infections, like Influenza A Virus. Since IFITM proteins facilitate PI3K signalling, is it possible that IFITM-derived peptides would inhibit cell growth and promote apoptosis? This might be desirable in the setting of cancer, but perhaps not in healthy individuals trying to stave off virus infections.

It seems conceivable that agents efficiently inhibiting SARS-CoV-2 infection of lung cells, cardiomyocytes and gut organoids might offer perspectives for therapy. Of course, there might be side-effects and it's a long way from basic research findings to application. Thus, we mention potential adverse effects (e.g. line 312-316) and present the findings with caution. However, some concerns of this reviewer seem rather far-fetched. Notably, the concern that such a treatment might increase susceptibility to other viruses doesn't really seem warranted because no long-term treatment is required. We added "in vitro" to the title and present potential therapeutic perspectives with caution.

Minor issues:

Line 291: text should be changed to "The antiviral activity of IFITMs is very broad and does not usually not involve interactions with specific viral glycoproteins."

Changed.

When citing reference 21, there are a host of other papers that could be cited. From 2014 to 2021, there are now several studies on the effects of IFITM proteins in virus-producing cells. I would recommend citing a few others.

We included an additional reference. Notably, Pubmed lists about 170 articles on the effect of IFITMs on viruses; thus, not all of them can be cited.

In Figure 6, the right side of 6A does not appear to be described in the Figure legend. What does “mIFITM2 peptide” signify? This reagent is not adequately described in the text, figure legend, or methods section. It also appears in Supplemental Figure 14 but is not described. The human-derived peptides are clearly described in Figure 5 but I’m not seeing a similar indication of what mIFITM2 is.

We now clarified that mIFITM2 refers to the commercially available mouse peptide that was used to generate the human version for further studies (revised legend to Figure 6).

Reviewer #2: the authors have addressed my concerns.

We are pleased that this reviewer had no remaining issues.

Reviewer #3: This is a revision of the manuscript by Bozzo CP et al. The authors have made extensive revision and added new data (and replicates) to address the potential concerns. In particular, new Fig. 4 e-g provides convincing proof of IFITM2-spike interactions and this likely takes place at the early endosomes.

We are pleased that we addressed most concerns of reviewer 3 and that he/she notes that our new data provide “convincing proof of IFITM2-spike interactions and this likely takes place at the early endosomes”.

However, this reviewer still has significant concerns over the flow cytometry data in Supplementary Fig. 10. Comparing panels a and d, The newly carried out IFN- β treatment altered the PE staining pattern, even for the isotype controls. IFN- β also did not seem to enhance intracellular IFITM2 levels. Does siRNA transfection lead to a specific reduction of surface IFITM2 in Calu3 cells? If the authors really believe that SARS-CoV-2 preferentially infects IFITM-positive cells (as is stated in the rebuttal letter), then such data should be added to the paper. Otherwise, I suggest simply removing the flow cytometry data to avoid confusion.

IFN- β had just marginal effects on PE staining of the isotype control (MFI isotype control Fig. S10b = 316, MFI Fig. S10e = 299). However, the previous figure was misleading since the axis was not presented in a biexponential scale as in Fig. S10b. We apologize for this inconsistency and corrected it. The raw data for permeabilized Calu-3 cells is shown in Fig R2. IFN- β treatment did not further enhance the percentage of IFITM2 expressing cells in total because >80% already stained positive in its absence, but clearly increased the levels of IFITM2 expression (see Fig. R2). Altogether these FACS results clearly show two important points: (1) IFITMs are accessible for inhibitory Ab binding and (2) IFN- β treatment increases surface-available IFITM2 and IFITM3 from about 20% to 60%. Thus, they are highly conclusive and contain important information. Consequently, we decided to keep them.

Figure R2: Raw mean fluorescence intensity (MFI) values for Fig. S10.